# DEPTRAI: DETACHABLE EXTERNAL-MEMORY LAYER FOR PARAMETER-TRANSFORMER INJECTION

## ABSTRACT

Large language models (LLMs) quickly become outdated because the factual knowledge they encode is fixed at training time, and retraining for every new fact is prohibitively expensive. Prior "internal" editors apply closed-form perturbations directly to the feed-forward weights, but each new patch is applied in place to the base model, causing edits to accumulate, interfere, and preventing straightforward revocation. We present DEPTRAI—**D**etachable **E**xternal-memory layer for **P**arameter-**Tra**nsformer **I**njection—that stores each edited fact as a key–value tuple outside the model, leaving all original weights frozen. At inference, the frozen FFN produces a subject key, which is routed to the nearest stored key using a Mahalanobis metric that mirrors the inverse-covariance scaling of closed-form editors. A lightweight gate then either substitutes the edited value or preserves the base projection. This design turns factual patching into a reversible database-style update rather than a permanent modification of parameters. DEPTRAI achieves the highest average performance on sequential editing tasks, outperforming the latest dual-memory method WISE by **15–20%**,

## 1 INTRODUCTION

Large language models (LLMs) such as Claude [1], Grok [2], and GPT-4 (Achiam et al., 2023) have shown a remarkable performance on many benchmarks (Achiam et al., 2023; Yang et al., 2024b; Wang et al., 2024). Their success is often attributed to an ability to encode an enormous amount of world knowledge directly in the parameters of a Transformer network, which makes them attractive as *implicit* knowledge bases (Feng et al., 2023; Delétang et al., 2023). However, implicit storage is a double-edged sword: models hallucinate, drift out of date, and resist fine-grained inspection. Retraining or fully fine-tuning an LLM after every factual change is prohibitively expensive, motivating the search for lightweight *knowledge-editing* techniques (De Cao et al., 2021; Jiang et al., 2024).

The dominant family of editing methods follows the *locate–then–edit* paradigm. Causal tracing first identifies the feed-forward (FFN) layers that mediate a target association, a low-rank perturbation $\Delta$ is then solved in closed form and added to the value matrix $W$ of those layers. ROME (Meng et al., 2022) demonstrates the approach for single facts, MEMIT (Meng et al., 2023) extends it to thousands of edits, and AlphaEdit (Fang et al., 2025) further constrains the update by projecting it onto the null space of preserved keys. Mathematically, all three methods boil down to a global Mahalanobis rescaling between the old output $Wk$ and the new slice $v$.

Dual-memory methods such as GRACE (Hartvigsen et al., 2023) and adapter-style memory such as WISE (Wang et al., 2025) address the entanglement of edits by attaching a side table of parameters and training a

---

[1] https://claude.ai/
[2] https://grok.com/

router that chooses between base model and patch. However, because the router relies on unwhitened cosine or dot-product similarity in the full hidden space, it remains sensitive to surface-form variation and carries a considerable memory footprint.

We introduce DEPTRAI (*Detachable External Parameter Transformer Retrieval and Injection*), a new editing framework that couples the precision of closed-form editors with the flexibility of an external memory. Building on ROME's insight that the last subject token alone addresses the factual association, we store each fact as a single subject key and its edited value in an external key–value table, leaving the original weights untouched.

At inference time the frozen FFN produces a query key $k$, DEPTRAI routes $k$ to the nearest stored key using a Mahalanobis distance that mirrors the inverse-covariance factor of MEMIT, then either substitutes the edited value or lets the base projection $Wk$ pass through.

Our main contributions.

- We present DEPTRAI, the *detachable* key–value memory that augments a frozen Transformer and realises knowledge edits without overwriting any internal parameter.
- We show that the Mahalanobis metric induced by the closed-form coefficient $\beta = \mu^\top C^{-1} k$ yields a principled router that is robust to surface-form variation, eliminating the key-bias of previous editors.
- Experiments on LLaMA 3.2-3B, Qwen 2.5 3B and LLaMA 3.1-8B across ZsRE, Hallucination show that DEPTRAI achieves the highest average score in sequential editing compared to recent methods such as WISE about 15-25%.

## 2 PRELIMINARIES

### 2.1 AUTOREGRESSIVE LANGUAGE MODELS AND MEMORY STORAGE

Large language models (LLMs) are typically trained in an autoregressive manner, predicting the next token $x_{[t]}$ based on the previous sequence of tokens $x_{[1]}, \ldots, x_{[t-1]}$. Formally, the conditional probability of the next token can be expressed as

$$x_{[t]} \mid x_{[1]}, \ldots, x_{[t-1]} \triangleq G([x_{[1]}, \ldots, x_{[t-1]}]) = \text{softmax}(W_y, h^D_{[t-1]}), \tag{1}$$

where $G$ denotes the transformer model (Vaswani et al., 2017), $W_y$ is the output embedding matrix, and $h^D_{[t-1]}$ is the hidden state at the final layer $D$ for the preceding token $x_{[t-1]}$.

The hidden states in the transformer are updated layer by layer using a combination of self-attention and feed-forward operations. Specifically, for a token $x$ at layer $l$, the hidden state $h^l$ is computed as

$$h^l = h^{l-1} + a^l + m^l, \tag{2}$$

where $a^l$ denotes the output of the multi-head attention module, and $m^l$ denotes the output of the feed-forward network (FFN). The feed-forward update $m^l$ is computed via

$$m^l = W^l_{\text{out}} \sigma(W^l_{\text{in}} \gamma(h^{l-1} + a^l)), \tag{3}$$

where $W^l_{\text{in}}$ and $W^l_{\text{out}}$ are learnable matrices, $\sigma$ is a non-linear activation function, and $\gamma$ denotes layer normalization (Ba et al., 2016).

Following the interpretations proposed in prior works (Bau et al., 2020; Meng et al., 2022), the FFN layers can be seen as a form of associative memory: the input $k$ (after applying $\sigma(W^l_{\text{in}}, \gamma(h^{l-1} + a^l))$) serves as a key, and the output $m^l$ serves as a value. This leads to the perspective that the weight matrix $W^l_{\text{out}}$ associates keys with corresponding stored values. Specifically,

$$m^l = W^l_{\text{out}} k, \tag{4}$$

where $k$ represents the intermediate key encoding derived from the hidden state and attention output.

Based on this understanding, most model editing methods focus on modifying the FFN layers to inject or update factual knowledge within LLMs. For clarity and consistency throughout the remainder of this paper, we denote $W$ as shorthand for $W_{\text{out}}^l$.

## 2.2 Model Editing in Large Language Models

Many studies have shown that LLMs inherently memorize a vast number of factual associations (Petroni et al., 2019; Brown et al., 2020; Chowdhery et al., 2023). Such knowledge is often expressed in the form of (subject, relation, object) triplets $(s, r, o)$, where a prompt like "Michael Jordan plays the sport of" triggers the model to predict "basketball".

The task of model editing concerns directly modifying factual associations encoded in the parameters of an LLM without retraining the entire model (Sinitsin et al., 2020; De Cao et al., 2021; Meng et al., 2022; 2023; Fang et al., 2025). Given a desired change in factual knowledge, editing methods aim to minimally update the model's parameters so that it consistently outputs the new fact while preserving unrelated knowledge.

Consider a list of desired edits, where $(s_i, r_i, o_i)$ are the subject, relation, and object of the $i$-th factual triplet. We assume there are no conflicting edits such that no two edits share the same $(s, r)$ but disagree on $o$. Each edit is associated with a prompt $p_i$ (e.g., "Michael Jordan plays the sport of") intended to elicit the new object $o_i$ (e.g., "football").

To implement such edits, most recent methods focus on updating the output weight matrices $W$ of the FFN layers. Suppose $W \in \mathbb{R}^{d_1 \times d_0}$, where $d_0$ and $d_1$ are the input and output dimensions of the FFN, respectively. For a set of $u$ edits, one constructs two matrices

$$K_1 = [k_1|k_2|\cdots|k_u] \in \mathbb{R}^{d_0 \times u}, \quad V_1 = [v_1|v_2|\cdots|v_u] \in \mathbb{R}^{d_1 \times u}, \tag{5}$$

where $k_i$ encodes $(s_i, r_i)$ and $v_i$ encodes $o_i$. The editing objective becomes minimizing the reconstruction error

$$\Delta = \underset{\tilde{\Delta}}{\arg\min} \, ||(W + \tilde{\Delta})K_1 - V_1||^2, \tag{6}$$

where $\Delta$ is the perturbation applied to $W$.

However, editing only based on new knowledge can cause catastrophic forgetting of unrelated memories (Gupta et al., 2024). To mitigate this, one typically incorporates an additional preservation term involving a matrix $K_0$ and $V_0$, representing the original keys and values from pre-existing knowledge

$$\Delta = \underset{\tilde{\Delta}}{\arg\min} \left( ||(W + \tilde{\Delta})K_1 - V_1||^2 + ||(W + \tilde{\Delta})K_0 - V_0||^2 \right). \tag{7}$$

Under the assumption that $WK_0 \approx V_0$ holds prior to editing (i.e., the model faithfully encodes the old knowledge), the optimal $\Delta$ can be derived using the normal equation (Lang, 2012)

$$\Delta = (V_1 - WK_1)K_1^T(K_0K_0^T + K_1K_1^T)^{-1}. \tag{8}$$

In practice, $K_0$ is approximated by collecting representations from a large corpus, typically using over 100,000 (subject, relation, object) triplets extracted from datasets like Wikipedia (?Fang et al., 2025). Despite being an approximation, this strategy enables scalable editing at the level of thousands of factual changes while maintaining fluency, generalization, and specificity (Meng et al., 2023).

In this work, we build upon these principles to propose improved editing mechanisms that better preserve old memories while ensuring effective assimilation of new information.

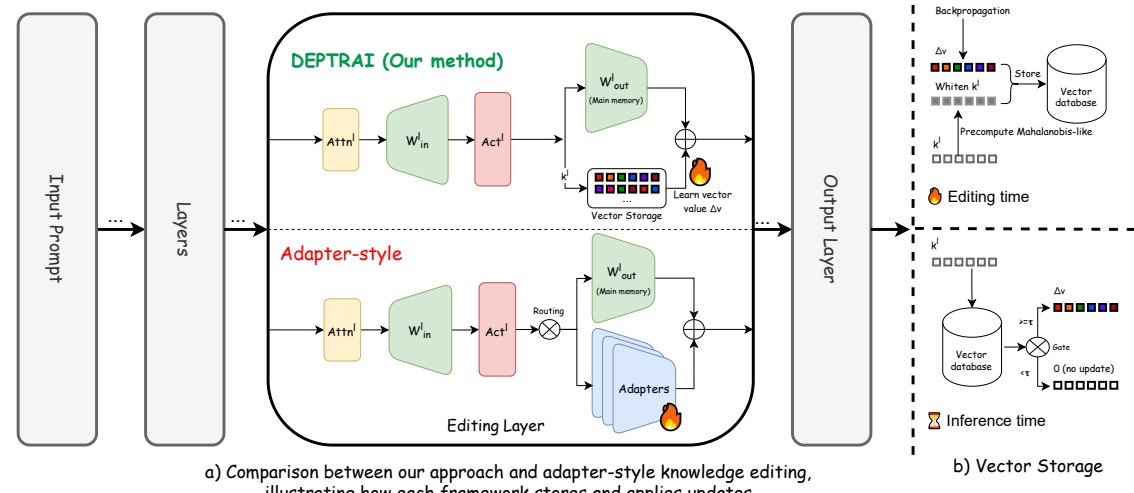

a) Comparison between our approach and adapter-style knowledge editing,
illustrating how each framework stores and applies updates

b) Vector Storage

Figure 1: Overview of DEPTRAI compared with adapter-style knowledge editing. (a) Adapter-based editors store updates as additional trainable modules injected into specific layers, requiring the model to route activations through these parameter "experts" during inference. In contrast, DEPTRAI detaches all edits from the backbone: each fact is stored as a whitened key and a learned value vector in an external vector database, and a Mahalanobis-style gate determines whether to substitute this edited value or fall back to the frozen FFN output. (b) Illustration of the DEPTRAI vector-storage mechanism, showing the whitening of subject keys, the learned value update $\Delta v$, and the gating decision that applies edits only when the key similarity exceeds a threshold.

## 3 METHODOLOGY

Unlike prior internal editors such as MEMIT and AlphaEdit, which inject low-rank weight perturbations and must carefully balance new and old knowledge. DEPTRAI leaves the base parameters untouched, stores each edited subject key and its value in a detachable external layer, and employs a Mahalanobis-based router to fetch the correct value at inference time, as shown in Figure 1

### 3.1 MOTIVATION

Starting from the closed-form update shared by MEMIT and AlphaEdit, the optimal perturbation to an FFN output matrix can be expressed as

$$\Delta = (V_1 - W K_1) K_1^\top C^{-1}, \tag{9}$$

where the covariance matrix

$$C = K_0 K_0^\top + K_1 K_1^\top \tag{10}$$

balances preserved keys $K_0$ and edited keys $K_1$. For a query key $k_i$, the edited layer output is

$$(W + \Delta)k_i = W k_i + (V_1 - W K_1) K_1^\top C^{-1} k_i. \tag{11}$$

Defining the mixing coefficient

$$\beta_i = K_1^\top C^{-1} k_i, \tag{12}$$

To connect this to our external-memory design in Figure 1, consider the *single-fact* case, where there is only one edited key–value pair, $k$ and $v$. Then $\beta_i$ becomes a scalar $\beta(k_i) = k^\top C^{-1} k_i$, and equation 11 simplifies

to

$$(W + \Delta)k_i = Wk_i + (v - Wk)k^\top C^{-1}k_i \tag{13}$$
$$= Wk_i + \beta(k_i)\,(v - Wk). \tag{14}$$

If we denote the base FFN output by $v_0 = Wk_i$ and the edit vector by $\Delta v = v - Wk$, we can write

$$(W + \Delta)k_i = v_0 + \beta(k_i)\,\Delta v. \tag{15}$$

Equation 15 makes the structure of the update explicit, the edited output is the original value $v_0$ plus a direction $\Delta v$ scaled by a data-dependent coefficient $\beta(k_i)$. In other words, the closed-form editor is already implementing a *gate* that interpolates between the base projection and the edited value.

AlphaEdit influences this gate indirectly by constraining $C^{-1}$ via a projection onto the null space of $K_0$. This acts as a coarse, global rescaling of $\beta(k_i)$, but it does not give fine-grained control over how strongly the edit should fire in different contexts.

DEPTRAI takes the single-fact perspective in equation 15 as a design blueprint. We externalize the key $k$ and the edit direction $\Delta v$ into a vector database, and interpret $\beta(k_i)$ as a Mahalanobis-style routing score. At inference time, the frozen FFN produces $v_0$ and a subject key; our external layer retrieves $\Delta v$ and applies an explicit gate $g(k_i)$ (derived from the same Mahalanobis geometry) to produce

$$v_0 + g(k_i)\,\Delta v, \tag{16}$$

as depicted in Figure 1. This view motivates DEPTRAI as a direct factorization of the closed-form update into a detachable codebook (holding keys and $\Delta v$) plus an interpretable gating mechanism.

## 3.2 EXTERNAL LAYER: DEPTRAI

**From in-place perturbation to detachable memory.** In MEMIT (Meng et al., 2023) and AlphaEdit (Fang et al., 2025), the closed-form update $\Delta$ is injected directly into the FFN output matrix $W$. This permanently modifies the base parameters and entangles edits over time. DEPTRAI instead keeps $W$ frozen and stores each factual edit in an explicit external memory table. This detachable structure allows facts to be added, revoked, or swapped at runtime, transforming editing into a lightweight retrieval-and-injection process.

**From mixing coefficient $\beta$ to Mahalanobis distance.** In the in-place closed-form formulation for a *single* edited fact with query key $k$, edited key $\mu$ and value $v$, the edited output becomes

$$(W + \Delta)k = v_0 + \beta(k)\,\Delta v. \tag{17}$$

where the mixing coefficient is the scalar

$$\beta(k) = \mu^\top C^{-1}k, \qquad C = C_0 + \mu\mu^\top. \tag{18}$$

Thus, $\beta(k)$ represents the projection of the query key $k$ onto the edited key $\mu$, scaled by the inverse covariance. With the whitening matrix $\Lambda = C^{-1} \succ 0$, the closed-form coefficient becomes

$$\beta(k) = \mu^\top \Lambda k. \tag{19}$$

The associated Mahalanobis distance is

$$d(k, \mu) = (k - \mu)^\top \Lambda (k - \mu) = \|k\|_\Lambda^2 + \|\mu\|_\Lambda^2 - 2\,\mu^\top \Lambda k. \tag{20}$$

Hence,

$$\beta(k) \;\propto\; -\,d(k, \mu), \tag{21}$$

up to terms independent of $\mu$. Maximizing $\beta(k)$ is therefore equivalent to minimizing the Mahalanobis distance between $k$ and the edited key. For efficiency, we precompute a whitened key

$$w = \Lambda\mu, \tag{22}$$

so that routing at inference reduces to the dot-product score

$$s(k) = w^\top k. \tag{23}$$

The full quadratic form never needs to be evaluated. This makes DEPTRAI's routing as cheap as a dot-product lookup while preserving the exact decision boundary implied by the Mahalanobis geometry of the closed-form editor.

**Memory structure.** For each factual edit $j$, we begin with a subject key $k_j \in \mathbb{R}^{d_{\text{in}}}$ and an edited value vector $v_j \in \mathbb{R}^{d_{\text{out}}}$, both extracted following the MEMIT procedure. As discussed the above section, the closed-form update for a single edit depends on a covariance term of the form

$$C_i = C_0 + k_i k_i^\top, \tag{24}$$

where the global covariance

$$C_0 \approx \mathbb{E}_x\big[k(x)k(x)^\top\big] \tag{25}$$

captures the geometry of preserved keys, and the rank-1 addition $k_i k_i^\top$ introduces the contribution of the new fact. DEPTRAI follows this formulation but adopts a lightweight storage. For each edit, we compute the inverse $C_i^{-1}$ only once during preprocessing in order to produce a Mahalanobis-whitened key. The matrix $C_i$ and its inverse are *not* stored. Instead, we retain only the transformed vector:

$$\mu_j = k_j, \qquad w_j = C_i^{-1}\mu_j. \tag{26}$$

This ensures that all routing decisions operate in the same Mahalanobis geometry induced by the closed-form solution, while eliminating the overhead of maintaining per-edit covariance matrices.

The external memory is serialized compactly as

$$\mathcal{E} = \big\{\,(\mu_j, w_j, v_j)\,\big\}_{j=1}^M, \tag{27}$$

where $\mu_j$ is the raw subject key, $w_j$ is its whitened counterpart used for similarity scoring, and $v_j$ is the edited value vector applied when the route is activated. Because only $(\mu_j, w_j, v_j)$ are stored, lookup at inference reduces to a dot-product search over whitened keys, avoiding any matrix inversion or dynamic recomputation.

**Routing rule.** Given a batch of queries $\{k_n\}_{n=1}^N$, DEPTRAI computes scores and selects the best-scoring fact

$$s_{nj} = w_j^\top k_n, \tag{28}$$

$$j_n^\star = \arg\max_j s_{nj}, \tag{29}$$

and activates an external addition if the score passes a similarity threshold $\tau$:

$$\text{gate}_n = \mathbf{1}\big[\, s_{nj_n^\star} \geq \tau \,\big]. \tag{30}$$

## 4 EXPERIMENTS

### 4.1 EXPERIMENTAL SETUP

We briefly introduce the evaluation metrics, datasets, and baseline methods. For more detailed descriptions of the experimental settings, please refer to Appendix B.

**Base LLMs & Baselines.** We run experiments on three LLMs: LLaMA 3.2-3B (Meta, 2024), Qwen 2.5-3B (Yang et al., 2024a), and LLaMA 3.1-8B (Meta, 2024). We compare DEPTRAI against parameter-editing baselines such as Fine-Tuning FT-L (Meng et al., 2022) and FT-M (Zhang et al., 2024), ELDER (Li et al., 2025), ROME (Meng et al., 2022), MEMIT (Meng et al., 2023), AlphaEdit (Fang et al., 2025). To assess sequential editing task, we compare our method with three long-life model editing methods: GRACE (Hartvigsen et al., 2023), and WISE (Wang et al., 2025).

**Datasets.** To evaluate sequential editing, we use the closed-book QA dataset ZsRE (Levy et al., 2017) and assess Hallucination correction on SelfCheckGPT (Manakul et al., 2023) following (Hartvigsen et al., 2023; Wang et al., 2025). For the single-edit setting, we adopt KnowEdit (Zhang et al., 2024) and report results on four selective tasks: CounterFact, ZsRE, WikiBio, and ConvSent (Appendix F).

**Metrics.** In sequential editing task, each edit $t \in \{1, \ldots, T\}$ provides an edit query $\mathbf{x}_e^t$ with target $\mathbf{y}_e^t$, optional paraphrases $\mathcal{X}_{e'}^t$ for testing generalization, and an unrelated statements $\mathcal{X}_{loc}^t$ for testing locality. Given an editing set $\mathcal{D}_{edit} = \{(\mathcal{X}_e^t, \mathcal{Y}_e^t)\}_{t=1}^T$, we evaluate the post-edit model $f_{\Theta_T}$ *after* all $T$ edits have been applied.

$$\mathbf{Rel.} = \frac{1}{T}\sum_{t=1}^T \mathbf{1}(f_{\Theta_T}(\mathbf{x}_e^t) = \mathbf{y}_e^t), \ \mathbf{Gen.} = \frac{1}{T}\sum_{t=1}^T \mathbf{1}(f_{\Theta_T}(\mathbf{x}_{e'}^t) = \mathbf{y}_e^t), \ \mathbf{Loc.} = \frac{1}{T}\sum_{t=1}^T \mathbf{1}(f_{\Theta_T}(\mathbf{x}_{loc}^t) = f_{\Theta_0}(\mathbf{x}_{loc}^t)) \quad (31)$$

where $\mathbf{1}(\cdot)$ denote the indicator function. We report mean scores across edits for reliability (**Rel.**), generalization (**Gen.**), and locality (**Loc.**). When paraphrases or locality probes contain multiple instances, we average within each set. Following (Hartvigsen et al., 2023; Wang et al., 2025), we assess locality on the Hallucination dataset using perplexity (PPL) and omit a generalization score due to the lack of a suitable metric.

To further evaluate the preservation of the LLMs' intrinsic knowledge after editing, we follow (Fang et al., 2025), evaluating on the General Capbility Tests before and after editing $T = 1000$ and $T = 5000$ samples from ZsRE Appendix C.

Table 1: Main sequential editing results on ZsRE (QA setting). $T$: number of sequential edits. Rel., Gen., Loc., and Avg. denote Reliability, Generalization, Locality, and Average. The results are highlighted as best, and second-best within a 15% margin of the best. For $T = 1$, we only highlight our ability to achieve the highest performance.

| Method | Model | T = 1 | | | | T = 10 | | | | T = 100 | | | | T = 1000 | | | |
|---|---|---|---|---|---|---|---|---|---|---|---|---|---|---|---|---|---|
| | | Rel.↑ | Gen.↑ | Loc.↑ | Avg.↑ | Rel.↑ | Gen.↑ | Loc.↑ | Avg.↑ | Rel.↑ | Gen.↑ | Loc.↑ | Avg.↑ | Rel.↑ | Gen.↑ | Loc.↑ | Avg.↑ |
| FT-L | LLaMA-3.2-3B | 100 | 100 | 100 | 100 | 48.00 | 46.00 | 75.70 | 56.57 | 32.85 | 27.00 | 23.00 | 27.62 | 16.35 | 12.60 | 3.00 | 10.65 |
| ELDER | | 100 | 100 | 100 | 100 | 88.00 | 68.50 | 80.10 | 78.87 | 65.15 | 52.77 | 66.23 | 61.38 | 59.67 | 47.86 | 48.12 | 51.88 |
| AlphaEdit | | 100 | 100 | 100 | 100 | 87.67 | 91.00 | 94.79 | 91.15 | 62.88 | 56.83 | 33.36 | 51.02 | 0.03 | 0.00 | 4.94 | 1.66 |
| GRACE | | 0.00 | 0.00 | 100 | 33.33 | 55.17 | 0.00 | 100 | 51.72 | 34.60 | 0.10 | 100 | 44.90 | 32.85 | 0.14 | 100 | 44.33 |
| WISE | | 100 | 100 | 100 | 100 | 71.83 | 70.16 | 100 | 80.66 | 60.87 | 57.37 | 99.73 | 72.66 | 57.69 | 55.64 | 99.63 | 70.99 |
| **DEPTRAI** | | 100 | 100 | 100 | 100 | 100 | 90.50 | 100 | 96.83 | 89.16 | 77.07 | 100 | 88.74 | 88.12 | 74.72 | 99.15 | 87.30 |
| FT-L | Qwen 2.5-3B | 100 | 100 | 100 | 100 | 49.50 | 47.83 | 80.29 | 59.21 | 30.88 | 29.41 | 26.29 | 28.86 | 15.69 | 13.14 | 3.00 | 29.18 |
| ELDER | | 100 | 90.00 | 100.0 | 96.67 | 83.00 | 81.50 | 82.75 | 82.42 | 58.52 | 48.45 | 70.67 | 59.21 | 51.18 | 44.02 | 54.57 | 49.92 |
| AlphaEdit | | 100 | 100 | 100 | 100 | 93.00 | 90.50 | 98.00 | 93.83 | 87.14 | 77.75 | 74.36 | 79.75 | 71.86 | 67.28 | 26.68 | 55.27 |
| GRACE | | 25.00 | 0.00 | 100 | 41.67 | 56.50 | 0.00 | 100 | 52.17 | 35.64 | 0.00 | 100 | 45.21 | 33.49 | 1.89 | 100 | 45.13 |
| WISE | | 100 | 100 | 100 | 100 | 49.50 | 48.50 | 100 | 66 | 49.37 | 45.00 | 100 | 64.79 | 44.52 | 42.44 | 100 | 62.32 |
| **DEPTRAI** | | 100 | 100 | 100 | 100 | 88.00 | 84.00 | 100 | 90.67 | 76.30 | 67.90 | 88.55 | 77.58 | 73.67 | 86.60 | 66.24 | 75.50 |
| FT-L | LLaMA-3.1-8B | 100 | 100 | 100 | 100 | 52.80 | 51.50 | 76.70 | 60.33 | 37.60 | 29.83 | 25.50 | 30.98 | 21.69 | 23.67 | 2.15 | 15.84 |
| ELDER | | 100 | 100 | 100 | 100 | 88.83 | 72.17 | 84.46 | 81.82 | 62.82 | 50.44 | 73.60 | 62.29 | 48.62 | 39.43 | 23.29 | 37.11 |
| AlphaEdit | | 100 | 100 | 100 | 100 | 85.17 | 80.67 | 81.29 | 82.38 | 58.59 | 54.36 | 22.68 | 45.21 | 2.91 | 2.71 | 3.00 | 2.87 |
| GRACE | | 0.00 | 0.00 | 100 | 33.33 | 52.66 | 0.00 | 100 | 50.89 | 34.73 | 1.23 | 100 | 45.32 | 31.96 | 1.38 | 100 | 44.45 |
| WISE | | 100 | 100 | 100 | 100 | 83.83 | 78.83 | 100 | 87.55 | 70.99 | 66.00 | 100 | 79.00 | 63.12 | 60.22 | 98.95 | 74.10 |
| **DEPTRAI** | | 100 | 100 | 100 | 100 | 100 | 87.50 | 100 | 95.83 | 91.38 | 80.26 | 100 | 90.55 | 93.50 | 79.35 | 100 | 90.95 |

Table 2: Main sequential editing results on Hallucination Dataset. $T$: number of sequential edits. Rel., Gen., Loc., and Avg. denote Reliability, Generalization, Locality, and Average, respectively. The results are highlighted as best, and second-best. For $T = 1$, we only highlight our ability to achieve the highest performance.

| Method | Model | $T = 1$ | | $T = 10$ | | $T = 100$ | | $T = 600$ | |
| --- | --- | --- | --- | --- | --- | --- | --- | --- | --- |
| | | Rel. (PPL↓) | Loc.↑ | Rel. (PPL↓) | Loc.↑ | Rel. (PPL↓) | Loc.↑ | Rel. (PPL↓) | Loc.↑ |
| FT-L | LLaMA-3.2-3B | 1.00 | 100 | 1.12 | 88.76 | 12.45 | 35.45 | 254.3 | 0.10 |
| AlphaEdit | | 1.59 | 88.09 | 3.16 | 88.69 | 128.0 | 0.20 | 249.7 | 0.10 |
| GRACE | | 4.96 | 100 | 14.64 | 100 | 16.26 | 100 | 41.48 | 100 |
| WISE | | 1.00 | 100 | 1.13 | 96.06 | 1.64 | 99.49 | 33.05 | 68.04 |
| **DEPTRAI** | | 1.00 | 100 | 1.17 | 100 | 7.88 | 99.93 | 35.33 | 99.57 |
| FT-L | Qwen 2.5-3B | 1.23 | 100 | 18.54 | 45.68 | 62.83 | 0.00 | 88.34 | 0.00 |
| AlphaEdit | | 1.10 | 100 | 1.38 | 96.1 | 9.17 | 87.94 | 182.7 | 50.38 |
| GRACE | | 5.52 | 100 | 14.53 | 100 | 29.31 | 100 | 134.5 | 100 |
| WISE | | 1.12 | 100 | 16.00 | 54.39 | 32.8 | 15.85 | 36.19 | 17.73 |
| **DEPTRAI** | | 1.04 | 100 | 1.37 | 100 | 15.53 | 99.12 | 36.31 | 99.16 |
| FT-L | LLaMA-3.1-8B | 1.00 | 100 | 6.78 | 67.89 | 14.67 | 32.65 | 315.6 | 0.00 |
| AlphaEdit | | 1.00 | 100 | 2.78 | 97.80 | 248.34 | 0.10 | 325.7 | 0.00 |
| GRACE | | 4.58 | 100 | 15.97 | 100 | 16.76 | 100 | 33.04 | 100 |
| WISE | | 1.01 | 100 | 1.62 | 96.72 | 2.00 | 99.74 | 25.31 | 95.04 |
| **DEPTRAI** | | 1.00 | 100 | 1.41 | 100 | 5.48 | 100 | 30.65 | 99.96 |

## 4.2 MAIN RESULT

As shown in Table 1, DEPTRAI remains stable under long edit sequences. Across all three base models, reliability remains at $100\%$ at $T = 10$ and stays above $88\%$ at $T = 1000$; generalization and locality are also among the top results. This suggests the edits are integrated without broader behavioral drift. By comparison, WISE degrades by roughly 15–20% at higher depths, AlphaEdit and FT-L drop sharply beyond $T = 10$, and GRACE maintains locality but at the expense of reliability and generalization.

To further evaluate large-scale behavior, we take the methods that still retain reasonable overall performance at $T = 1000$, specifically DEPTRAI, WISE, ELDER and extend the evaluation to much longer horizons. Figure 2 presents results for $T = 2000$–$5000$, showing that DEPTRAI continues to sustain high reliability and locality even under thousands of edits, whereas baseline methods degrade steadily as the edit depth increases. Notably, WISE preserves locality well on the Qwen2.5-3B model, but its reliability and generalization still decline substantially with larger edit streams.

Table 2 indicates that DEPTRAI is consistently the most stable approach on the hallucination benchmark: it maintains near-perfect locality ($\approx 100\%$) across depths and ranks first or second on the reliability proxy (lower PPL is better) from $T{=}1$ through $T{=}600$ for all three base models. WISE is competitive at small $T$ and often second-best, but its PPL grows more noticeably as edits accumulate. GRACE preserves locality by design (frequently 100%) yet does so with substantially higher PPL, reflecting weaker reliability under sequential edits. Fine-tuning baselines (FT-L) and AlphaEdit degrade quickly with depth—PPL rises and locality erodes—highlighting how naive or overly broad edits can bleed into unrelated contexts. Overall, DEPTRAI achieves the best robustness profile: edits remain localized while reliability holds up even under long edit chains.

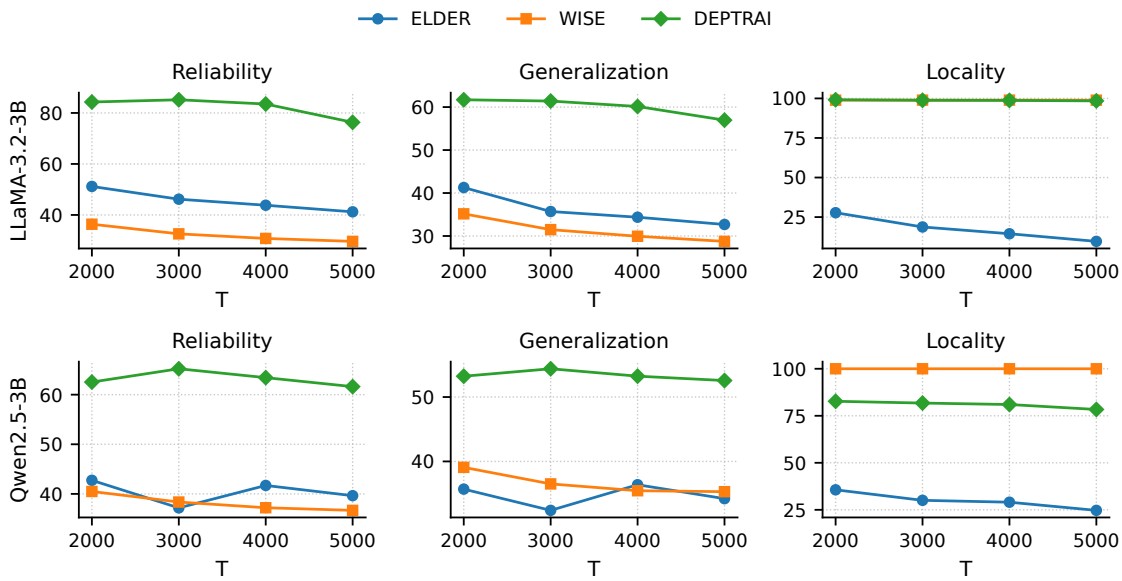

Figure 2: Sequential editing performance at large edit depths ($T = 2000$–$5000$). Across both LLaMA-3.2-3B and Qwen-2.5-3B, DEPTRAI sustains high reliability and locality, whereas ELDER and WISE show increasing degradation as the number of edits grows.

## 5 RELATED WORKS

### 5.1 LEVERAGING EXTERNAL KNOWLEDGE

External knowledge can be injected without retraining by retrieving demonstrations or memory entries. MemPrompt (Madaan et al., 2022) augments prompts with user feedback, while IKE (Zheng et al., 2023) leverages diverse demonstrations (copy, update, retrain) for reliable fact editing. Yet such methods often lack ripple effects: inserting one fact does not propagate to its implications (Cohen et al., 2024). To address this, decomposition-based editors (Zhong et al., 2023; Gu et al., 2024; Wang et al., 2025) break large edits into sequential sub-edits, while others combine counterfactual knowledge with classifiers to decide when to invoke the edited model (Mitchell et al., 2022b).

### 5.2 EXTENDING HIDDEN STATES

Another line of work modifies hidden representations directly, reducing the need for long prompts. Patching methods interpolate new and old hidden states to steer model outputs (Murty et al., 2022). Others augment FFN states with additional neurons (Dong et al., 2022; Huang et al., 2023) or use LoRA-style low-rank adapters to inject knowledge (Wu et al., 2023; Yu et al., 2024; Biderman et al., 2024). REMEDI (Hernandez et al., 2024) incorporates attribute vectors for entities, while GRACE (Hartvigsen et al., 2023) maintains a dynamic codebook of updates.

### 5.3 EDITING INTERNAL PARAMETERS

Finally, parameter-editing methods directly alter weights. Hypernetwork-based approaches predict $\Delta W$ for each edit (Sinitsin et al., 2020; Han et al., 2023; Tan et al., 2024), including KE (De Cao et al., 2021) and

SLAG (Hase et al., 2023), but are costly at scale. MEND (Mitchell et al., 2022a) improves efficiency via rank-one decomposition. Other works use causal tracing to locate critical hidden states for more targeted edits (Meng et al., 2022; 2023). To limit side effects, AlphaEdit (Fang et al., 2025) projects perturbations into the null space of preserved keys.

# 6 CONCLUSION

We introduced DEPTRAI, a detachable external-memory layer that edits LLMs without modifying base weights. By reinterpreting the closed-form mixing coefficient of internal editors as a Mahalanobis-style routing rule in key space, DEPTRAI turns factual patching into a reversible, database-like retrieval–and–injection step. Across LLaMA-3.2-3B, Qwen2.5-3B, LLaMA-3.1-8B, and Qwen3-8B, DEPTRAI sustains high reliability and near-perfect locality over long edit sequences (up to 5,000 edits), and on ZsRE sequential editing it consistently outperforms recent dual-memory baselines such as GRACE, or Adapter-style memory such as WISE and ELDER by roughly 15–20% at depth, while preserving general capabilities on GLUE, MMLU, GSM8K, AIME, and IFEval.

At the same time, our design exposes several limitations. First, DEPTRAI inherits the geometry of the underlying model: when the subject-key space is well structured (e.g., LLaMA-3 and Qwen3-8B), routing is clean and stable, but for noisier representations (as observed in Qwen2.5-3B) locality and some capability scores can degrade under very long edit sequences. Second, the current key–value construction is optimized for single-hop factual relations; on harder KnowEdit-style portability tests involving abbreviations, transliterations, or multi-hop reasoning, both our method and prior editors remain brittle, indicating that the bottleneck is the base representation and the value vector $v_j$, not only the routing metric. Finally, DEPTRAI still relies on a hand-tuned global threshold for activation, chosen from the Mahalanobis score distribution; we do not yet adapt this threshold online as the memory grows or as the edit stream changes.

These limitations suggest several directions for future work: shaping key representations during pre-training or instruction tuning to better support external routing; learning richer value encoders that more faithfully capture the intended edit; developing dynamic or learned thresholds for long-horizon lifelong editing; and combining detachable codebook-style memory with fine-tuning–based editors when large, coherent batches of edits are available. We hope DEPTRAI and our analysis of Mahalanobis routing help clarify both the promise and the boundaries of train-free, external-memory approaches to model editing.

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

## A  DETAIL OF DEPTRAI

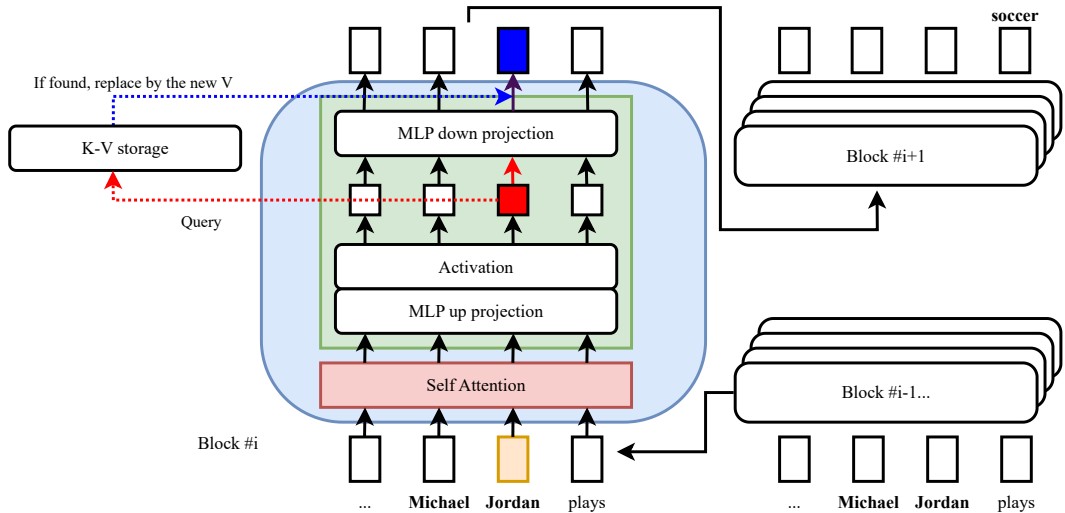

Figure 3: Detailed flow of DEPTRAI inside a transformer block. The self-attention and MLP down-projection produce a query key, which is compared against external K–V storage. If a match is found, the stored value $V$ is injected through the MLP up-projection, replacing the original activation (e.g., updating "Michael Jordan plays basketball" → "Michael Jordan plays soccer"); otherwise, the original pathway is preserved.

Figure 3 illustrates the detailed inference flow of DEPTRAI within a transformer block. When processing an input sequence, each block proceeds through the standard self-attention and activation steps. Afterward, the MLP down-projection generates a query key vector for the current subject token.

DEPTRAI introduces an external detachable key–value (K–V) storage module. During inference, the generated query key is routed against the stored keys using a Mahalanobis-based distance metric. If a close match is found, the associated edited value $V$ is retrieved and substituted into the up-projection path of the MLP, effectively overriding the original factual association. If no suitable match exists, the model simply forwards the unaltered hidden representation through the up-projection.

This design enables DEPTRAI to (i) preserve the base model parameters intact, (ii) inject or update knowledge through explicit K–V entries, and (iii) flexibly add, clear, or swap edits at runtime without retraining. The flow ensures that factual corrections, such as replacing "Michael Jordan plays basketball" with "Michael Jordan plays soccer," propagate seamlessly through subsequent layers while retaining locality and fluency.

## B IMPLEMENTATION DETAILS

### B.1 DESCRIPTIONS OF COMPARED METHODS

**FT-L** (Meng et al., 2022). We freeze the LLM except for a *single* MLP layer, which we fine-tune with an autoregressive loss. An $\ell_\infty$ constraint keeps the updated parameters close to the pretrained weights to limit drift.

**FT-M** (Zhang et al., 2024). It trains the same FFN layer as FT-L using the cross-entropy loss on the target answer while masking the original text

**ROME** (Meng et al., 2022). A closed-form editor that identifies the MLP layer most responsible for a fact and applies a least-squares update to its weight matrix to implant the new relation in one shot.

**MEMIT** (Meng et al., 2023). A multi-layer extension of ROME that performs coordinated, closed-form updates across several MLP layers, enabling efficient batch or large-scale injections of facts while minimizing side effects.

**AlphaEdit** (Fang et al., 2025). An optimization-based editor that learns a compact parameter delta to satisfy the edited outputs under locality-preserving regularization, yielding strong reliability with controlled collateral change.

**GRACE** (Hartvigsen et al., 2023). A lifelong editor that maintains a discrete key–value codebook of edits. At inference, it retrieves the nearest key to the current input and, when appropriate, replaces intermediate activations, thereby isolating new knowledge from the base model.

**WISE** (Wang et al., 2025). A long-horizon editing method that combines locality-aware training with a selection mechanism to preserve earlier edits; it remains more reliable than simple fine-tuning under many sequential edits while avoiding excessive drift.

### B.2 TRAINING DETAILS AND HYPERPARAMETERS

**General setup.** We evaluate on three base models: LLaMA 3.2-3B, Qwen 2.5-3B, and LLaMA 3.1-8B. Batch size is 1 for sequential editing. All runs use 4×NVIDIA A100 40GB GPUs (each experiment is reproducible on a single A100).

**Layer selection.** For **MEMIT**, and **AlphaEdit** (internal parameter editing), we target the mid–upper MLP band: layers $[4, 5, 6, 7, 8]$. For **ROME**, **DEPTRAI** (ours), we edit *layer 8* on all three models.

**FT-L (single-layer fine-tuning).** We use the public ROME codebase[3]. LR grid $\{1e-5, 1e-4, 5e-4\}$, 50 steps; we report the best at $5e-4$. All other weights are frozen, and we apply an $\ell_\infty$ constraint to limit drift.

**FT-M (multi-layer fine-tuning).** A stronger FT baseline that updates a small stack of adjacent transformer blocks (same LR grid as FT-L) and early stopping keyed to locality. This typically yields higher ES but increases interference risk.

**GRACE.** We follow the released setup: LR = 1.0, and `replace_last` (replace only last-token activations in AR decoding). We lightly sweep $\epsilon_{\text{init}}$ for stability; other knobs remain at defaults.

**WISE.** We use the authors' suggested settings and evaluate strictly in *retrieve* mode, *without replay and without merging*. Optimization uses SGD (Shamir & Zhang, 2013) with LR = 1.0 for LLaMA 3.2-3B and Qwen 2.5-3B, and LR = 0.9 for LLaMA 3.1-8B. During editing we set $\rho$=0.2 and routing thresholds $\alpha$=5.0, $\beta$=20.0, $\gamma$=10.0 (for LLaMA 3.1-8B we use $\alpha$=2.0, $\beta$=20.0, $\gamma$=10.0). We use

---

[3] https://github.com/kmeng01/rome

`n_iter=70`, `act_ratio=0.88` for LLaMA 3.2-3B and Qwen 2.5-3B; `n_iter=30`, `act_ratio=0.50` for LLaMA 3.1-8B; `norm_constraint=1.0` and `objective=only_label` for all. The edited parameter is the MLP down-projection of a single layer per model: *layer 20* for LLaMA 3.2-3B, *layer 23* for Qwen 2.5-3B, and *layer 29* for LLaMA 3.1-8B. No merging/sharding stage is applied; edits are applied via retrieval-time routing only.

**DEPTRAI (ours).** We construct the edit vector $V^*$ by following the MEMIT work. The resulting delta is written at *layer 8* for LLaMA 3.2-3B, Qwen 2.5-3B, and LLaMA 3.1-8B. At inference, routing uses a Mahalanobis score on the layer-8 activation with model-specific thresholds: $\tau=0.4$ (LLaMA 3.2-3B), $\tau=0.6$ (Qwen 2.5-3B), and $\tau=0.5$ (LLaMA 3.1-8B).

## C  GENERAL CAPABILITY

Table 3: F1 scores (%) on GLUE tasks and MMLU after **1000 sequential edits** ZsRE.

| Model | Setting | SST | MRPC | RTE | CoLA | NLI | MMLU | Avg. |
|---|---|---|---|---|---|---|---|---|
| LLaMA 3.2-3B | Pre-edited | 95.49 | 58.94 | 29.26 | 55.9 | 68.44 | 55.43 | 60.58 |
| | Post-edited | 95.49 | 58.94 | 29.26 | 55.9 | 68.44 | 55.43 | 60.58 |
| | $\Delta$ | 0.00 | 0.00 | 0.00 | 0.00 | 0.00 | 0.00 | 0.00 |
| Qwen 2.5-3B | Pre-edited | 94.49 | 69.99 | 18.72 | 76.98 | 76.76 | 59.96 | 66.15 |
| | Post-edited | 95.00 | 67.99 | 16.99 | 74.00 | 77.68 | 58.37 | 65.01 |
| | $\Delta$ | ↑0.51 | ↓0.2 | ↓1.73 | ↓2.98 | ↑0.92 | ↓1.59 | ↓1.32 |
| LLaMA 3.1-8B | Post-edited | 95.99 | 64.28 | 24.5 | 78.69 | 73.26 | 59.25 | 66.00 |
| | Pre-edited | 95.99 | 64.28 | 24.5 | 78.69 | 73.26 | 58.65 | 65.90 |
| | $\Delta$ | 0.00 | 0.00 | 0.00 | 0.00 | 0.00 | ↓0.6 | 0.1 |

1. **SST (Stanford Sentiment Treebank)** (Socher et al., 2013): single-sentence sentiment classification on movie-review sentences with human-annotated binary labels.
2. **MRPC (Microsoft Research Paraphrase Corpus)** (Dolan & Brockett, 2005): sentence-pair classification to determine whether two sentences are semantically equivalent.
3. **MMLU (Massive Multi-Task Language Understanding)** (Hendrycks et al., 2021): a broad knowledge and reasoning evaluation measuring multi-task accuracy.
4. **RTE (Recognizing Textual Entailment)** (Bentivogli et al., 2009): natural language inference determining whether a premise logically entails a hypothesis.
5. **CoLA (Corpus of Linguistic Acceptability)** (Warstadt et al., 2019): single-sentence classification of grammatical acceptability.
6. **NLI (Natural Language Inference)** (Williams et al., 2018): inference over sentence pairs to identify their logical relationship.

Although GLUE and MMLU provide broad measurements of linguistic competence, they are primarily classification or multiple-choice evaluations. Such formats indicate whether the post-edited model preserves recognition-based skills, but they do not fully capture generative reasoning or multi-step compositional abilities. To better assess the model's behavior after large-scale knowledge updates, we therefore include several free-form generation benchmarks— GSM8K (Cobbe et al., 2021), AIME'24/'25 - American Invitational Mathematics Examination, MATH500 (Lightman et al.), SimpleQA (Wei et al., 2024), and IFEval (Zhou et al., 2023) —which require arithmetic reasoning, symbolic manipulation, or coherent instruction following. These tasks are known to be far more sensitive to internal disruptions introduced by sequential edits.

As reported in Table 3, DEPTRAI largely preserves the general capabilities of the underlying models after undergoing 1000 sequential ZsRE edits. Both LLaMA-3.2-3B and LLaMA-3.1-8B exhibit virtually no degradation across the GLUE tasks or MMLU, with average differences below 0.1 F1. This indicates that extensive factual editing does not compromise their broader linguistic or reasoning skills.

Table 4: AIME, MATH500, GSM8K, SimpleQA, and IFEval performance across three models. We report pre-edit, post-edit, and the change $\Delta$.

| Task / Subtask | LLaMA-3.2-3B | | | Qwen2.5-3B | | | LLaMA-3.1-8B | | |
|---|---|---|---|---|---|---|---|---|---|
| | Pre (%) | Post (%) | $\Delta$ | Pre (%) | Post (%) | $\Delta$ | Pre (%) | Post (%) | $\Delta$ |
| **AIME, MATH500, GSM8K, SimpleQA** | | | | | | | | | |
| AIME'24 | 0.0 | 0.0 | 0.0 | 0.0 | 0.0 | 0.0 | 0.0 | 0.0 | 0.0 |
| AIME'25 | 0.0 | 0.0 | 0.0 | 3.33 | 0.0 | $-3.33$ | 0.0 | 0.0 | 0.0 |
| MATH500 | 7.4 | 7.4 | 0.0 | 18.4 | 3.0 | $-15.4$ | 12.2 | 12.2 | 0.0 |
| GSM8K | 26.0 | 26.0 | 0.0 | 14.0 | 28.4 | $+14.4$ | 51.4 | 51.4 | 0.0 |
| SimpleQA | 3.33 | 3.33 | 0.0 | 2.50 | 0.76 | $-1.74$ | 4.44 | 4.42 | $-0.02$ |
| **IFEval Instruction Adherence** | | | | | | | | | |
| Prompt Strict | 6.2 | 6.2 | 0.0 | 22.8 | 14.6 | $-8.2$ | 9.2 | 9.2 | 0.0 |
| Inst Strict | 11.0 | 11.0 | 0.0 | 31.95 | 26.52 | $-5.43$ | 13.45 | 13.45 | 0.0 |
| Prompt Loose | 7.0 | 7.0 | 0.0 | 24.0 | 15.8 | $-8.2$ | 11.0 | 11.0 | 0.0 |
| Inst Loose | 11.51 | 11.51 | 0.0 | 33.64 | 27.43 | $-6.21$ | 15.01 | 15.01 | 0.0 |

A similar trend appears in the broader capability benchmarks summarized in Table 4 (AIME'24/'25, MATH500, GSM8K, SimpleQA, and IFEval). For both LLaMA backbones, post-edit performance remains almost unchanged, reflecting strong robustness to large volumes of injected knowledge.

However, Qwen-2.5-3B behaves differently. While some tasks remain stable, others—particularly MATH500, SimpleQA, and IFEval—show non-trivial declines after sequential edits. This contrast suggests that DEPTRAI's robustness may depend on properties of the underlying model family, such as representational geometry or layer-wise key alignment. To investigate this further, we conduct additional experiments on a larger and more recent model, Qwen3-8B (Appendix D), to examine whether the observed sensitivity persists in stronger backbones.

## D EXPERIMENTS ON QWEN3-8B

The degradation observed in Qwen-2.5–3B does not appear in the larger and more recent Qwen3-8B model. As shown in Table 7 and Figure 4, Qwen3-8B maintains strong reliability and near-perfect locality even at extreme edit depths (T=2000–5000), with performance curves closely tracking those of the LLaMA series. Its GLUE and MMLU scores remain unchanged after 1000 edits (Table 6), and its generative reasoning tasks—AIME'24/'25, MATH500, GSM8K, SimpleQA, and IFEval—show effectively no degradation in either standard or reasoning-enabled modes.

This contrast suggests that the performance decline observed earlier in Qwen 2.5–3B is not inherent to DEPTRAI, but rather model-dependent. We hypothesize that the weaker robustness of Qwen 2.5-3B arises from properties of its training pipeline or pretraining dataset, which may yield a noisier or less stable key-space geometry. Since DEPTRAI relies on high-quality subject-key representations to populate its external vector memory, insufficiently structured or inconsistent internal keys can propagate noise into the stored value vectors, ultimately degrading downstream general-capability tasks.

The stable results of Qwen3-8B, despite undergoing the same sequential edits, support this hypothesis, its improved architecture and training corpus appear to produce cleaner, more consistent key representations, leading to durable performance across all metrics. These findings highlight an important consideration for external-memory editing methods, the underlying model's representation geometry plays a crucial role in determining long-horizon editing robustness.

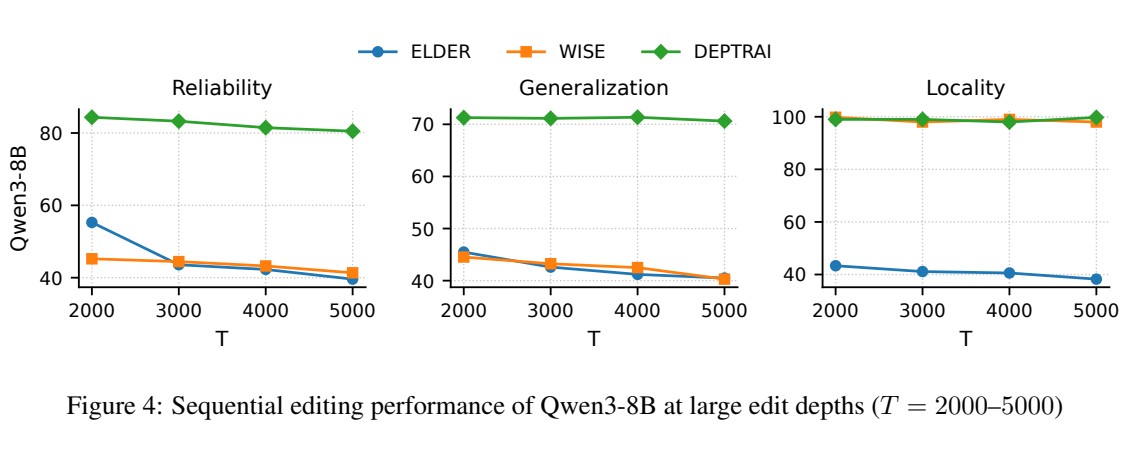

Figure 4: Sequential editing performance of Qwen3-8B at large edit depths ($T = 2000$–$5000$)

Table 5: Main sequential editing results on ZsRE (QA setting) for Qwen3-8B. $T$: number of sequential edits. Rel., Gen., Loc., and Avg. denote Reliability, Generalization, Locality, and Average. The results are highlighted as best , and second-best within a 15% margin. For $T = 1$, only our method is highlighted.

| Method | Model | $T = 1$ | | | | $T = 10$ | | | | $T = 100$ | | | | $T = 1000$ | | | |
|---|---|---|---|---|---|---|---|---|---|---|---|---|---|---|---|---|---|
| | | Rel.↑ | Gen.↑ | Loc.↑ | Avg.↑ | Rel.↑ | Gen.↑ | Loc.↑ | Avg.↑ | Rel.↑ | Gen.↑ | Loc.↑ | Avg.↑ | Rel.↑ | Gen.↑ | Loc.↑ | Avg.↑ |
| ELDER | Qwen3-8B | 100.00 | 100.00 | 100.00 | 100.00 | 95.50 | 89.50 | 92.50 | 92.50 | 72.05 | 63.30 | 73.58 | 69.64 | 64.60 | 54.50 | 50.24 | 56.45 |
| WISE | | 100.00 | 100.00 | 100.00 | 100.00 | 84.50 | 79.50 | 100.00 | 88.00 | 72.41 | 68.89 | 100.00 | 80.43 | 58.51 | 55.11 | 97.66 | 70.43 |
| DEPTRAI | | 100.00 | 100.00 | 100.00 | 100.00 | 100.00 | 79.50 | 100.00 | 93.17 | 86.81 | 76.16 | 100.00 | 87.66 | 86.22 | 72.62 | 99.74 | 86.19 |

# E  BATCH EDIT EXPERIMENTS

Although our context is designed for sequential, sample-by-sample updates, we also evaluate the batch editing setting to better understand the behavior of different editing paradigms. Sequential editing represents the most challenging regime for parameter-based editors. Each update alters the model weights, so later edits must operate on parameters that have already been modified by earlier ones. This accumulation of interference is precisely what causes degradation in methods such as MEMIT or AlphaEdit when $T$ becomes large.

Batch editing serves as an informative control condition. When all $T$ edits are applied simultaneously, parameter-editing methods can harmonize the updates in a single closed-form solve, avoiding the compounding drift that arises in sequential mode. Comparing sequential and batch performance therefore allows us to isolate the origin of degradation whether it stems from the editor itself or from the iterative accumulation of weight perturbations.

From this perspective, batch results serve two purposes. First, they reveal how well parameter-based editors behave when interference is removed, showing the upper bound of their performance. Second, they highlight the contrast with DEPTRAI's external-memory design, because DEPTRAI never modifies the base weights, its performance does not depend on batching and remains stable across all $T$. The results from Table 8 validate that routing-based editing intrinsically avoids the interference problem that batch editing is designed to mitigate.

# F  KNOWEDIT BENCHMARK

Table 9 shows high edit success (ES) across methods but comparatively lower portability (Port.) and locality (Loc.), largely due to KnowEdit's stress design. Portability is tested with aliases, synonyms, paraphrases, and

Table 6: MMLU and GLUE benchmark (F1 score) for Qwen3-8B. We report pre-edit, post-edit, and the difference $\Delta$.

| Task | Pre (%) | Post (%) | $\Delta$ |
|------|---------|----------|----------|
| MMLU | 71.74 | 71.74 | 0.00 |
| SST | 96.50 | 96.50 | 0.00 |
| MRPC | 75.28 | 74.75 | $-0.53$ |
| RTE | 18.79 | 18.79 | 0.00 |
| CoLA | 80.00 | 80.00 | 0.00 |
| NLI | 85.76 | 85.76 | 0.00 |

Table 7: Comparison of Qwen3-8B in Non-reasoning mode (chat template) and Reasoning mode (chat template, enable_thinking=True). We report pre-edit, post-edit, and the difference $\Delta$.

| Task / Subtask | Non-reasoning | | | Reasoning | | |
|----------------|---------|----------|----------|---------|----------|----------|
| | Pre (%) | Post (%) | $\Delta$ | Pre (%) | Post (%) | $\Delta$ |
| **AIME, MATH500, GSM8K, SimpleQA** | | | | | | |
| AIME24 | 28.10 | 28.10 | 0.00 | 73.33 | 73.33 | 0.00 |
| AIME25 | 21.33 | 21.33 | 0.00 | 66.67 | 66.67 | 0.00 |
| MATH500 | 85.20 | 85.20 | 0.00 | 95.40 | 95.40 | 0.00 |
| GSM8K | 93.25 | 93.25 | 0.00 | 94.20 | 94.20 | 0.00 |
| SimpleQA | 2.61 | 2.59 | $-0.02$ | 2.54 | 2.54 | 0.00 |
| **IFEval Instruction Adherence** | | | | | | |
| Prompt Strict | 81.33 | 81.33 | 0.00 | 82.21 | 82.21 | 0.00 |
| Inst Strict | 87.41 | 87.41 | 0.00 | 88.33 | 88.33 | 0.00 |
| Prompt Loose | 85.21 | 85.21 | 0.00 | 86.93 | 86.93 | 0.00 |
| Inst Loose | 89.93 | 89.93 | 0.00 | 91.06 | 91.06 | 0.00 |

light compositions that intentionally differ from the edited surface form. When the learned edit binds too tightly to the original phrasing, transfer fails, leading to a drop in portability. Locality is probed by pairing the edited subject tokens with unrelated predicates or questions, a "near-miss" setup that routes the model toward the edited entity while requiring the pre-edit response, so edits that globally modulate the subject representation can bleed into these contexts and trigger false activations, lowering Loc. DEPTRAI generally offers the best balance, maintaining strong ES while limiting collateral effects. The benchmark construction makes Port. and Loc. harder than ES.

## G ABLATION STUDY

### G.1 COMPARISON WITH OTHER ROUTING METHODS

Furthermore, we experiment comparing Mahalanobis against cosine similarity for 3 models on the KnowEdit benchmark and report the performance in Table 10.

As depicted in Table 10, although the editing score (ES) of Mahalanobis is relatively lower compared to cosine similarity for 3 models, the portability (Port.), locality (Loc.), and fluency (F.) are significantly higher. We hypothesize that by using cosine similarity as the distance method, the set of key synonyms $\mathcal{K}$ could contain both actual synonyms and related phrases (e.g., a synonym of "dog" is "canine", but related phrases such

Table 8: Batch editing results for LLaMA-3.2-3B and Qwen2.5-3B.

| Model | Method | $T = 10$ | | | $T = 100$ | | | $T = 1000$ | | |
|---|---|---|---|---|---|---|---|---|---|---|
| | | Rel. | Gen. | Loc. | Rel. | Gen. | Loc. | Rel. | Gen. | Loc. |
| LLaMA-3.2-3B | MEMIT | 88.50 | 88.50 | 97.95 | 2.43 | 2.43 | 0.17 | 0.00 | 0.00 | 0.00 |
| | MEMIT-batch | 84.00 | 84.00 | 99.69 | 79.97 | 79.37 | 84.17 | 76.09 | 72.35 | 68.86 |
| | AlphaEdit | 87.67 | 91.00 | 94.79 | 62.88 | 56.83 | 33.36 | 0.03 | 0.00 | 4.94 |
| | AlphaEdit-batch | 85.17 | 82.67 | 93.99 | 77.82 | 73.72 | 73.98 | 67.26 | 64.76 | 50.76 |
| | DEPTRAI | 100.0 | 90.50 | 100.0 | 89.16 | 77.07 | 100.0 | 88.12 | 74.72 | 99.15 |
| Qwen2.5-3B | MEMIT | 84.50 | 71.00 | 53.08 | 0.00 | 0.00 | 0.00 | 0.00 | 0.00 | 0.00 |
| | MEMIT-batch | 83.00 | 75.50 | 83.33 | 78.36 | 76.53 | 76.53 | 88.38 | 23.69 | 17.48 |
| | AlphaEdit | 93.00 | 90.50 | 98.00 | 93.83 | 87.14 | 77.75 | 71.86 | 67.28 | 26.68 |
| | AlphaEdit-batch | 89.50 | 79.50 | 99.00 | 89.07 | 78.16 | 89.14 | 89.42 | 81.18 | 78.16 |
| | DEPTRAI | 88.00 | 84.00 | 100.0 | 76.30 | 67.90 | 88.55 | 73.67 | 86.60 | 66.24 |

Table 9: Performance across models and editing methods on the KnowEdit benchmark. ES = Edit Sucess, Port. = Portability, Loc. = Locality, and F. = Fluency. The results are highlighted as best , second-best , and third-best .

| Method | Model | ZsRE | | | | WikiBio | | | WikiCounterFact | | | | ConvSent | | |
|---|---|---|---|---|---|---|---|---|---|---|---|---|---|---|---|
| | | ES↑ | Port.↑ | Loc.↑ | F.↑ | ES↑ | Loc.↑ | F.↑ | ES↑ | Port.↑ | Loc.↑ | F.↑ | ES↑ | Loc.↓ | F.↑ |
| FT-L | LLaMA 3.2-3B | 52.71 | 45.69 | 67.43 | 350.51 | 66.54 | 57.82 | 589.76 | 46.50 | 39.59 | 49.96 | 434.81 | 51.82 | 0.00 | 494.86 |
| FT-M | | 100.00 | 58.36 | 86.41 | 395.03 | 100.00 | 91.60 | 602.73 | 100.00 | 72.77 | 69.82 | 508.61 | 47.57 | 0.00 | 464.77 |
| ROME | | 99.12 | 51.61 | 46.99 | 527.57 | 99.16 | 35.99 | 591.09 | 99.25 | 54.79 | 37.21 | 588.95 | 45.14 | 0.00 | 612.79 |
| MEMIT | | 97.18 | 51.21 | 51.67 | 522.18 | 87.85 | 70.54 | 627.58 | 97.08 | 51.79 | 39.88 | 579.10 | 46.57 | 0.00 | 588.80 |
| AlphaEdit | | 98.40 | 51.09 | 45.85 | 521.34 | 92.90 | 68.31 | 627.05 | 98.15 | 57.33 | 35.19 | 589.21 | 43.55 | 0.00 | 591.22 |
| DEPTRAI | | 97.15 | 51.61 | 41.22 | 521.73 | 94.78 | 67.42 | 621.55 | 98.64 | 59.20 | 33.87 | 577.68 | 42.85 | 0.00 | 584.83 |
| FT-L | Qwen 2.5-3B | 53.93 | 45.64 | 73.42 | 493.01 | 66.33 | 79.86 | 606.95 | 45.15 | 33.60 | 50.48 | 528.26 | 49.50 | 0.00 | 607.86 |
| FT-M | | 99.98 | 60.31 | 89.78 | 552.26 | 100.00 | 93.38 | 612.69 | 100.00 | 74.36 | 76.76 | 575.62 | 46.10 | 0.00 | 592.52 |
| ROME | | 96.77 | 52.63 | 53.67 | 573.75 | 96.08 | 62.74 | 617.69 | 98.57 | 55.92 | 51.97 | 584.04 | 45.79 | 0.00 | 606.32 |
| MEMIT | | 95.37 | 52.67 | 48.32 | 563.31 | 94.40 | 61.51 | 616.65 | 98.05 | 58.56 | 46.62 | 575.96 | 44.75 | 0.00 | 602.62 |
| AlphaEdit | | 97.18 | 53.50 | 49.32 | 580.00 | 91.5 | 67.45 | 617.69 | 99.2 | 45.03 | 46.64 | 598.28 | 43.5 | 0.00 | 612.28 |
| DEPTRAI | | 98.95 | 55.01 | 52.55 | 586.16 | 92.86 | 57.13 | 628.86 | 99.36 | 55.01 | 39.82 | 598.43 | 39.16 | 0.00 | 624.34 |
| FT-L | LLaMA 3.1-8B | 50.29 | 38.18 | 51.11 | 350.91 | 62.04 | 70.41 | 571.88 | 49.21 | 38.45 | 32.69 | 394.80 | 51.72 | 0.00 | 505.59 |
| FT-M | | 100.00 | 59.23 | 79.30 | 418.66 | 100.00 | 87.26 | 599.32 | 100.00 | 73.40 | 62.35 | 518.36 | 48.33 | 0.00 | 462.15 |
| ROME | | 98.91 | 52.41 | 48.48 | 551.85 | 91.49 | 66.66 | 627.68 | 99.19 | 57.33 | 40.77 | 591.05 | 44.88 | 0.00 | 608.20 |
| MEMIT | | 97.65 | 50.36 | 69.01 | 573.11 | 82.02 | 83.88 | 630.25 | 97.09 | 40.76 | 61.19 | 599.93 | 48.94 | 0.00 | 594.03 |
| AlphaEdit | | 84.81 | 48.75 | 77.38 | 579.29 | 90.43 | 67.09 | 628.00 | 82.54 | 34.35 | 69.95 | 605.63 | 41.89 | 0.00 | 594.80 |
| DEPTRAI | | 94.99 | 52.46 | 68.98 | 575.26 | 95.51 | 69.81 | 624.84 | 97.97 | 59.71 | 41.85 | 579.55 | 45.43 | 0.00 | 594.26 |

as "cat" or "kitty" also have close distances to "dog" under cosine similarity space). Therefore, the editing method could modify the incorrect set of keys, leading to poorer locality and portability scores compared to Mahalanobis. This phenomenon does not happen to Mahalanobis, as this distance first clusters semantically equivalent points, thus eliminating the related phrases in the first place.

Table 10: Performance across models and distance methods on the KnowEdit benchmark. Best results are shown in red.

| Model | Method | ZsRE | | | | WikiBio | | | WikiCounterFact | | | | ConvSent | | |
|---|---|---|---|---|---|---|---|---|---|---|---|---|---|---|---|
| | | ES↑ | Port.↑ | Loc.↓ | F.↑ | ES↑ | Loc.↑ | F.↑ | ES↑ | Port.↑ | Loc.↑ | F.↑ | ES↑ | Loc.↑ | F.↑ |
| LLaMA 3.2-3B | Mahalanobis | 97.15 | 51.61 | 41.22 | 521.73 | 94.78 | 67.42 | 621.55 | 98.64 | 59.20 | 33.87 | 577.68 | 42.85 | 0.00 | 584.83 |
| | Cosine 0.6 | 99.60 | 47.11 | 40.35 | 351.52 | 91.90 | 56.29 | 622.98 | 99.64 | 44.44 | 30.77 | 489.32 | 41.95 | 0.00 | 579.52 |
| Qwen2.5-3B | Mahalanobis | 98.95 | 55.01 | 52.55 | 586.16 | 92.86 | 57.13 | 628.86 | 99.36 | 55.01 | 39.82 | 598.43 | 39.16 | 0.00 | 624.34 |
| | Cosine 0.6 | 99.49 | 50.31 | 52.98 | 586.92 | 94.38 | 55.84 | 626.29 | 99.43 | 52.97 | 39.49 | 596.86 | 39.16 | 0.00 | 624.34 |
| LLaMA 3.1-8B | Mahalanobis | 94.99 | 52.46 | 68.98 | 575.26 | 95.51 | 69.81 | 624.84 | 97.97 | 59.71 | 41.85 | 579.55 | 45.43 | 0.00 | 594.26 |
| | Cosine 0.6 | 97.09 | 50.17 | 52.51 | 524.07 | 90.64 | 69.98 | 627.18 | 98.89 | 48.61 | 41.26 | 581.60 | 43.50 | 0.00 | 590.07 |

Table 11: Ablation of edit-layer selection for LLaMA-3.2-3B. We report Reliability (Rel.), Generalization (Gen.), and Locality (Loc.) at $T = 100$ and $T = 1000$. Best results are highlighted as best and second-best within 15% as second-best.

| Method | $T = 100$ | | | $T = 1000$ | | |
|---|---|---|---|---|---|---|
| | Rel. | Gen. | Loc. | Rel. | Gen. | Loc. |
| DEPTRAI-L4 | 85.62 | 71.57 | 100.0 | 87.64 | 72.35 | 98.51 |
| DEPTRAI-L8 | 89.16 | 77.07 | 100.0 | 88.12 | 74.72 | 99.15 |
| DEPTRAI-L15 | 82.22 | 63.89 | 94.33 | 65.47 | 48.95 | 76.69 |
| DEPTRAI-L25 | 90.22 | 59.28 | 54.84 | 81.75 | 52.86 | 44.98 |

## G.2 LAYER SELECTION

To determine the most effective intervention point for DEPTRAI, we follow the causal-tracing procedure introduced in ROME (Meng et al., 2022), which identifies layers that carry the strongest causal influence over subject–object factual associations. ROME's analysis consistently shows that mid-FFN layers encode subject-specific identity features more cleanly than shallow or deep layers. Motivated by this, we test multiple FFN layers in LLaMA-3.2-3B to validate whether DEPTRAI exhibits similar behavior. Table 11 reports Reliability, Generalization, and Locality for edits applied at Layers 4, 8, 15, and 25 under T=100 and T=1000. The trend mirrors the causal-tracing prediction: mid-layers, particularly Layer 8, achieve the best balance of edit accuracy and isolation.

## G.3 THRESHOLD SELECTION

To choose an appropriate routing threshold $\tau$, we compute the Mahalanobis similarity score for each subject token and its surrounding non-subject tokens across all edited records. Figure 5 displays these distributions for four different base models. In every case, subject tokens form a compact cluster at higher similarity values, whereas other tokens spread across lower scores. This separation is exactly the structure DEPTRAI relies on: a clean subject band enabling confident activation, and a dispersed non-subject region preventing accidental overrides.

Given these distributions, we define a feasible range for $\tau$ as any value lying strictly between the subject and non-subject clusters. Intuitively, a larger threshold biases the system toward preservation, since the router only fires when a subject key is extremely close to a stored fact; this minimizes false activations and maximizes locality. Conversely, a smaller threshold favors higher generalization, allowing the router to cover mild paraphrases or slightly mismatched subject mentions, at the potential cost of activating more often.

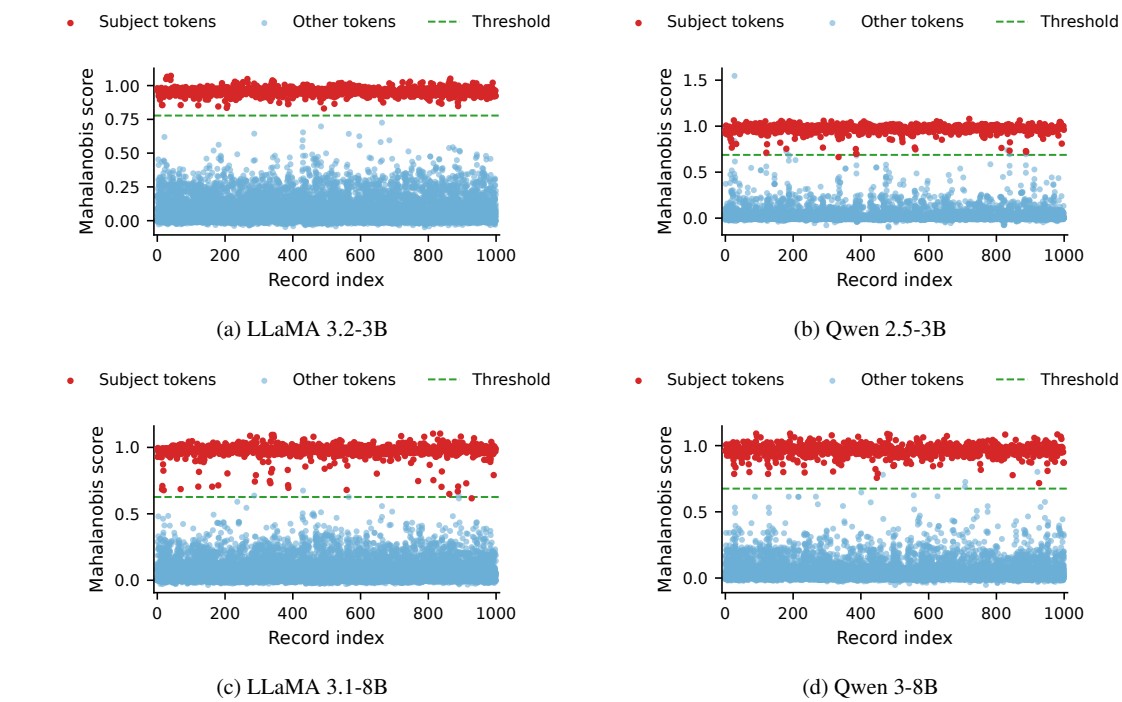

Figure 5: Mahalanobis scores across model variants.

Across the four models, the optimal threshold differs slightly due to variations in key-space geometry. LLaMA models exhibit a sharper subject–non-subject margin, enabling a wider safe range of $\tau$. Qwen2.5, however, shows a more entangled distribution—consistent with the performance drift observed in Section C—suggesting its key representation is less cleanly factorized and thus requires a more conservative threshold.

Overall, the separation patterns in Figure 5 provide a direct, data-driven method for choosing the edit-activation threshold, balancing locality and generalization depending on deployment needs.

