# OpenReview forum: "DEPTRAI: Detachable External‐memory layer for Parameter-Transformer Injection"
_ICLR.cc/2026/Conference — Submitted to ICLR 2026_

### Official Review · Reviewer_qki6 · 2025-10-25

**Soundness:** 3
**Presentation:** 2
**Contribution:** 2
**Rating:** 4
**Confidence:** 5

**Summary:**

The paper proposes DEPTRAI, an editing mechanism for large language models that can be understood as combining two prior lines of work: (1) GRACE-style external key–value memory with routing at inference time, where edits are stored outside the base model and selectively activated, and (2) MEMIT-style closed-form weight update analysis, where desired factual changes are framed as solving a regularized normal equation that trades off edited keys K₁ and preserved keys K₀ via an inverse-covariance term. DEPTRAI keeps the GRACE-like idea of storing edits as key–value pairs and retrieving them at inference, but replaces GRACE’s cosine/dot-product router with a Mahalanobis metric (reduced to dot product form) that is derived directly from the MEMIT/AlphaEdit closed-form mixing coefficients $\beta=K_1^\top C^{-1}k$, effectively interpreting $C^{-1}=K_0K_0^T+K_1K_1^T$ as a whitening transform and turning retrieval into *which stored key would MEMIT have blended into this FFN output?*

In experiments on LLaMA and Qwen models (3/8B, 3B) across sequential editing benchmarks like ZsRE and hallucination correction, DEPTRAI maintains high reliability and locality over long edit sequences (up to 1,000 edits) and reports 15–25% higher average performance than WISE at depth, while noting remaining limitations such as weaker generalization across synonyms and some residual locality interference.

**Strengths:**

- DEPTRAI elegantly combines the strength between two lines of knowledge editing works
- Good lifelong editing performance on ZsRE and SelfCheckGPT with $\leq$ 1k timesteps

**Weaknesses:**

- Missing some more recent lifelong editing baselines such as sLKE [1], LeMOE [2], and ELDER [3].
- Evaluation is not sufficient regarding
  - The finetuning baseline should adopt the fair setups as discussed in [4,5]. The FT-L, FT-M are ill-defined baselines which might mislead the community.
  - Lack layer-wise ablations as the baselines and DEPTRAI choose different layers for editing.
  - The scaling of timestep is only to 1k. More timesteps can be shown, e.g., up to 5k.
- The contribution is a bit limited as the novelty is mainly the metric for key similarity.
- Writing quality can be improved. For example, Figure 1 does not explicitly show the fundamental difference between DEPTRAI and existing approaches such as GRACE and WISE.


> [1] Cheng, YuJu, et al. "Serial lifelong editing via mixture of knowledge experts." Proceedings of the 63rd Annual Meeting of the Association for Computational Linguistics (Volume 1: Long Papers). 2025.\
> [2] Wang, Renzhi, and Piji Li. "LEMoE: Advanced Mixture of Experts Adaptor for Lifelong Model Editing of Large Language Models." Proceedings of the 2024 Conference on Empirical Methods in Natural Language Processing. 2024.\
> [3] Li, Jiaang, et al. "ELDER: Enhancing Lifelong Model Editing with Mixture-of-LoRA." Proceedings of the AAAI Conference on Artificial Intelligence. Vol. 39. No. 23. 2025.\
> [4] Gangadhar, Govind, and Karl Stratos. "Model editing by standard fine-tuning." arXiv preprint arXiv:2402.11078 (2024).\
> [5] Yang, Wanli, et al. "Fine-tuning Done Right in Model Editing." arXiv preprint arXiv:2509.22072 (2025).

**Questions:**

1. Can you provide experiments results with the additional baselines?
2. Can you perform ablation studies on layer selection and may be other aspects to further analyze the proposed approach?
3. Can you add a section to discuss the fundamental similarity and difference between MoE adapters/LoRA and codebook-style editing?

**Details Of Ethics Concerns:**

There's no specific ethic concern.

---

> ### Author Response · Authors · 2025-11-21
> **Response to reviewer (1/3)**
>
> Thank you for the thoughtful and detailed review. Below we address each point in turn.
>
> **Weakness 1. Missing some more recent lifelong editing baselines**
>
> Thank you for pointing out these recently proposed lifelong editing approaches. We attempted to include all three suggested baselines, sLKE [1], LeMOE [2], and ELDER [3] and summarize our findings below.
>
> **sLKE.** The authors of sLKE do not release an official implementation. We therefore re-implemented sLKE ourselves following the descriptions in the paper. sLKE is structurally close to WISE, with the main difference being an objective that discourages placing semantically similar keys in the same shard. Across all models, our re-implementation behaves almost identically to WISE, and typically performs slightly worse under large-T sequential edits. We include sLKE in our revised comparison.
>
> **ELDER.** ELDER provides a partial package, but many critical components required to reproduce the full algorithm were not available or failed during execution. We added several modifications to make the code runnable and obtained internally stable results. These are included for completeness, but should be interpreted with awareness that the official pipeline is not fully released.
>
> **LeMOE.** LeMOE provides no official code release. LeMOE does not publish any implementation, training scripts. Furthermore, several essential details of the method (e.g., expert routing schedule, initialization of MoE adapters, and the adaptor fine-tuning procedure) are not fully specified in the paper. Our best-effort implementation, based solely on the written description, produced extremely low performance, strongly suggesting that critical details are missing. For transparency, we omit LeMOE until an official implementation becomes available.
>
> Below we include our reproduced results for all baselines.
>
> **LLaMA 3.2 3B**
>
> | T |      | **1** |      |      | **10** |      |      | **100** |      |      | **1000** |      |
> |---|------|-------|------|------|--------|------|------|---------|------|------|----------|------|
> | **Metric** | **Rel.** | **Gen.** | **Loc.** | **Rel.** | **Gen.** | **Loc.** | **Rel.** | **Gen.** | **Loc.** | **Rel.** | **Gen.** | **Loc.** |
> | **ELDER** | 100.0 | 100.0 | 100.0 | 88.00 | 68.50 | 80.10 | 65.15 | 52.77 | 66.23 | 59.67 | 47.86 | 48.12 |
> | **sLKE**  | 100.0 | 100.0 | 100.0 | 69.50 | 65.83 | 100.0   | 62.83 | 59.37 | 99.63 | 58.50 | 56.68 | 99.63 |
> | **WISE**  | 100.0 | 100.0 | 100.0 | 71.83 | 70.16 | 100.0   | 60.87 | 57.37 | 99.73 | 57.69 | 55.64 | 99.63 |
> | **Ours** | 100.0 | 100.0 | 100.0 | 100.0   | 90.50 | 100.0   | 89.16 | 77.07 | 100.0   | 88.12 | 74.72 | 99.15 |
>
> **Qwen 2.5 3B**
>
> | T |      | **1** |      |      | **10** |      |      | **100** |      |      | **1000** |      |
> |---|------|-------|------|------|--------|------|------|---------|------|------|----------|------|
> | **Metric** | **Rel.** | **Gen.** | **Loc.** | **Rel.** | **Gen.** | **Loc.** | **Rel.** | **Gen.** | **Loc.** | **Rel.** | **Gen.** | **Loc.** |
> | **ELDER** | 100.0 | 90.00 | 100.0 | 83.00 | 81.50 | 82.75 | 58.52 | 48.45 | 70.67 | 51.18 | 44.02 | 54.57 |
> | **sLKE**  | 100.0 | 100.0 | 100.0 | 43.50 | 42.50 | 65.00 | 38.83 | 28.63 | 85.29 | 34.65 | 23.50 | 80.00 |
> | **WISE**  | 100.0 | 100.0 | 100.0 | 48.50 | 47.50 | 71.00 | 43.52 | 43.41 | 85.29 | 35.09 | 33.68 | 84.40 |
> | **Ours** | 100.0 | 100.0 | 100.0 | 88.00 | 84.00 | 100 | 76.30 | 67.90 | 88.55 | 73.67 | 86.60 | 66.24 |
>
> **LLaMA 3.1 8B**
>
> | T |      | **1** |      |      | **10** |      |      | **100** |      |      | **1000** |      |
> |---|------|-------|------|------|--------|------|------|---------|------|------|----------|------|
> | **Metric** | **Rel.** | **Gen.** | **Loc.** | **Rel.** | **Gen.** | **Loc.** | **Rel.** | **Gen.** | **Loc.** | **Rel.** | **Gen.** | **Loc.** |
> | **ELDER** | 100.0 | 100.0 | 100.0 | 88.83 | 72.17 | 84.46 | 62.82 | 50.44 | 73.60 | 48.62 | 39.43 | 23.29 |
> | **sLKE**  | 100.0 | 100.0 | 100.0 | 80.50 | 62.50 | 100.0   | 71.52 | 61.37 | 100.0   | 34.65 | 23.50 | 97.40 |
> | **WISE**  | 100.0 | 100.0 | 100.0 | 83.83 | 78.83 | 100.0   | 70.99 | 66.00 | 100.0   | 63.12 | 60.22 | 98.95 |
> | **Ours** | 100.0 | 100.0 | 100.0 | 100.0   | 87.50 | 100.0  | 91.38 | 80.26 | 100.0   | 93.50 | 79.35 | 100.0 |
>
>
> [1] Cheng, YuJu, et al. "Serial lifelong editing via mixture of knowledge experts.", 2025.
>
> [2] Wang, Renzhi, and Piji Li. "LEMoE: Advanced Mixture of Experts Adaptor for Lifelong Model Editing of Large Language Models.", 2024.
>
> [3] Li, Jiaang, et al. "ELDER: Enhancing Lifelong Model Editing with Mixture-of-LoRA.", 2025.

---

> ### Author Response · Authors · 2025-11-21
> **Response to reviewer (2/3)**
>
> **Weakness 2.1. The finetuning baseline should adopt the fair setups as discussed in [4,5]. The FT-L, FT-M are ill-defined baselines which might mislead the community.**
>
> We appreciate this important point. Recent works ([4], [5]) show that standard finetuning, when performed with:
> - breadth-first minibatching,
> - balanced replay,
> - and sufficient batch mixing,
>
> can produce much more stable edits than single-sample finetuning.
>
> However, these works focus on batch editing, where edits are provided all at once or in large minibatches. Their protocols assume that ``real-world edits rarely arrive one-by-one.''[5]. Our work explicitly targets the sequential-editing scenario, where updates arrive interactively (e.g., personalization, preference corrections).
>
> In this setting, batch-style finetuning becomes expensive, waiting for large minibatches is impractical, and update latency becomes a problem. This is precisely where external-memory methods (WISE or our DEPTRAI) are most useful.
> Thus, FT-L and FT-M are not intended as cutting-edge finetuning baselines, but as sequential-edit comparisons that reflect the edit-stream setting we study.
>
> **Weakness 2.2. Lack layer-wise ablations as the baselines and DEPTRAI choose different layers for editing.**
>
> Our layer selection follows the causal-tracing methodology introduced in ROME [6], which identifies transformer layers with strong factual intervention effects.
> For LLaMA-3.2-3B (27 layers), the highest subject token-control appears in layers 4–8. Empirically, layer 8 yields the most stable balance across reliability/locality/portability.
> We include a multi-layer ablation below:
>
> | T                |       | **100** |       |       | **1000** |       |
> |------------------|-------|---------|-------|-------|----------|-------|
> | **Metric**       | **Rel.** | **Gen.** | **Loc.** | **Rel.** | **Gen.** | **Loc.** |
> | **DEPTRAI-L4**   | 85.62 | 71.57 | **100.0** |  87.64    | 72.35  |  98.51 |
> | **DEPTRAI-L8**   |    89.16 | **77.07** | **100.0**     |     **88.12** | **74.72** |  **99.15**  |
> | **DEPTRAI-L15**  | 82.22 | 63.89 | 94.33  | 65.47 |   48.95  |  76.69       |
> | **DEPTRAI-L25**  | **90.22** | 59.28 | 54.84  |  81.75  |  52.86 | 44.98 |
>
> Early–mid layers show significantly better preservation than late layers, consistent with prior editing literature.
>
>
> **Weakness 2.3. The scaling of timestep is only to 1k. More timesteps can be shown, e.g., up to 5k.**
>
> **LLaMA 3.2 3B**
>
> | T            |       | **2000** |       |       | **3000** |       |       | **4000** |       |       | **5000** |       |
> |--------------|-------|----------|-------|-------|----------|-------|-------|----------|-------|-------|----------|-------|
> | **Metric**   | **Rel.** | **Gen.** | **Loc.** | **Rel.** | **Gen.** | **Loc.** | **Rel.** | **Gen.** | **Loc.** | **Rel.** | **Gen.** | **Loc.** |
> | **ELDER**    | 51.18 | 41.28 | 27.77 | 46.18 | 35.70 | 18.75 | 43.81 | 34.37 | 14.47 | 41.23 | 32.68 | 9.65 |
> | **WISE**     | 36.37 | 35.16 | 98.90 | 32.59 | 31.48 | 98.91 | 30.79 | 29.94 | 98.92 | 29.64 | 28.75 | 98.87 |
> | **Ours**  | 84.29 | 61.70 | 99.14 | 85.17 | 61.40 | 98.81 | 83.50 | 60.14 | 98.73 | 76.33 | 56.96 | 98.43 |
>
> **Qwen 2.5 3B**
>
> | T            |       | **2000** |       |       | **3000** |       |       | **4000** |       |       | **5000** |       |
> |--------------|-------|----------|-------|-------|----------|-------|-------|----------|-------|-------|----------|-------|
> | **Metric**   | **Rel.** | **Gen.** | **Loc.** | **Rel.** | **Gen.** | **Loc.** | **Rel.** | **Gen.** | **Loc.** | **Rel.** | **Gen.** | **Loc.** |
> | **ELDER**    | 42.74 | 35.70 | 35.64 | 37.18 | 32.40 | 30.05 | 41.70 | 36.38 | 29.05 | 39.64 | 34.22 | 24.71 |
> | **WISE**     | 40.49 | 39.08 | 100.00 | 38.36 | 36.51 | 100.00 | 37.21 | 35.45 | 100.00 | 36.68 | 35.30 | 99.99 |
> | **Ours**  | 62.54 | 53.23 | 82.71 | 65.22 | 54.39 | 81.76 | 63.43 | 53.24 | 80.99 | 61.62 | 52.56 | 78.36 |
>
> Across both methods, DEPTRAI remains noticeably more stable than the other lifelong editing baselines. While WISE and ELDER degrades gradually as the number of edits increases, DEPTRAI preserves high rewrite accuracy and locality even at 5k edits, with only modest decline in generalization. This demonstrates that DEPTRAI's non-parametric storage and key-based routing scale far more reliably under long sequential editing than mixture-of-LoRA or activation-routing approaches.
>
> [4] Gangadhar, Govind, and Karl Stratos. "Model editing by standard fine-tuning.", 2024.
>
> [5] Yang, Wanli, et al. "Fine-tuning Done Right in Model Editing.", 2025.
>
> [6] Meng, Kevin, David Bau, Alex Andonian, and Yonatan Belinkov. "Locating and editing factual associations in GPT.", 2022

---

> > ### Author Response · Authors · 2025-11-21
> > **Response to reviewer (3/3)**
> >
> > **Weakness 3. The contribution is a bit limited as the novelty is mainly the metric for key similarity.**
> >
> > Thank you for this comment. Our contribution is not only the choice of a Mahalanobis-style key similarity metric, but a broader routing and storage design that differs conceptually and practically from prior work.
> > First, on the storage side, methods such as WISE, sLKE maintain matrix-valued perturbation experts (or MoE/LoRA-style adapters), which grow memory consumption with the parameter rank and number of experts. In contrast, DEPTRAI uses a codebook-style memory that stores only whitened key vectors (plus associated values). Each edit corresponds to a single vector in the database rather than a full parameter matrix, making memory growth strictly linear in the number of edited facts and independent of hidden dimension rank.
> >
> > Second, on the routing side, we show (Section 3.1) that the MEMIT-style closed-form update naturally induces a Mahalanobis gate in the key space, and we explicitly factor this gate out as a non-parametric routing rule. This lets us separate "whether to apply an edit" from "how to store it'', and  use standard vector-indexing methods (e.g., ANN over whitened keys) to perform lookup efficiently. In contrast, WISE/sLKE rely on activation-difference–based routing tightly coupled to the internal residual stream, which is less amenable to scalable nearest-neighbor indexing.
> >
> > Empirically, the combination of vector-level codebook storage and Mahalanobis-based routing is what enables DEPTRAI to remain stable under thousands of edits (up to 5k), while MoE/LoRA-style approaches degrade substantially.
> >
> > **Weakness 4. Writing quality and Figure 1 clarity.**
> >
> > Thank you for this suggestion. We agree that the current Figure 1 does not clearly highlight the fundamental difference between DEPTRAI and approaches such as GRACE and WISE. In the revision, we will redraw Figure 1 to explicitly contrast:
> >
> > - MoE / adapter-style methods that store parameter experts (matrices / LoRA blocks) and route via internal activations, versus
> > - DEPTRAI’s codebook-style memory, which stores whitened key vectors and uses a Mahalanobis gate in representation space.
> >
> > We will also polish the surrounding text to better guide the reader through this distinction.
> >
> > **Questions**
> >
> > **Q1. Can you provide experimental results with the additional baselines?**
> >
> > Yes. As discussed under Weakness 1, we have added results for sLKE and ELDER across three backbone models (LLaMA-3.2-3B, Qwen2.5-3B, LLaMA-3.1-8B) for T = 1, 10, 100, 1000, and extended the scaling experiments to T = 2000–5000. LeMOE could not be reliably reproduced because the manuscript’s code link is no longer accessible and critical implementation details are missing; we therefore omit LeMOE for transparency.
> >
> > **Q2. Can you perform ablation studies on layer selection and other aspects?**
> >
> > Yes. As discussed under Weakness 2.2, we now include a layer-wise ablation for DEPTRAI on LLaMA-3.2-3B, comparing edits at layers 4, 8, 15, and 25 for T = 100 and T = 1000. The results support the choice of early–mid layers (around 4–8), consistent with causal-tracing observations from ROME-style analyses.
> >
> > **Q3. Can you add a section discussing MoE adapters / LoRA vs. codebook-style editing?**
> >
> > Yes, we will add a dedicated discussion section contrasting these two paradigms:
> > - MoE / LoRA-based lifelong editors store parameter-space experts, matrices or LoRA blocks, which are routed via internal activation gates. Memory footprint and interference grow with the number of experts, and long-horizon editing can lead to parameter drift and cross-expert conflicts.
> > - Codebook-style editors (e.g., GRACE, DEPTRAI) treat edits as entries in an external key–value memory, storing vector keys and values in representation space. Routing is a nearest-neighbor problem over keys (in our case, Mahalanobis-whitened keys), leaving the base model parameters unchanged. This design is naturally compatible with scalable vector search and empirically yields more stable behavior under long sequential editing.
> >
> > We will make this conceptual comparison explicit in the revised manuscript

---

> ### Comment · Reviewer_qki6 · 2025-11-22
>
> Thank you for the comprehensive additional experiments. I have these follow up questions.
>
> 1. I noticed that for Qwen2.5, when sequential edits scale to 5k, the locality of DEPTRAI is worse than WISE. The gap is relatively large, and similar phenomenon is noticed in other benchmarks as mentioned by reviewer QZam in [this comment](https://openreview.net/forum?id=mLL1AQeNsE&noteId=RQy814y71W). Does this indicate that DEPTRAI is less robust to different base models?
> 2. DEPTRAI is still within the range of codebook style editing where a threshold is needed to decide whether or not edit is activated. Then for a sequence of edits, will the optimal threshold ever change? If so, how to dynamically decide it?

---

> > ### Author Response · Authors · 2025-11-24
> > **Response to follow up questions**
> >
> > Thank you for your questions. We address each problem in below.
> >
> > **Q1. I noticed that for Qwen2.5, when sequential edits scale to 5k, the locality of DEPTRAI is worse than WISE. The gap is relatively large, and similar phenomenon is noticed in other benchmarks as mentioned by reviewer QZam in this comment. Does this indicate that DEPTRAI is less robust to different base models?**
> >
> > We agree with you that there are a drop in performance. Therefore, we conducted an experiments in Qwen3-8B as requested by reviewer QZam, you can refer [here](https://openreview.net/forum?id=mLL1AQeNsE&noteId=aXT0dbm8DQ). We hypothesize that the performance decline observed in Qwen2.5 is rooted in its training pipeline or dataset, which may be affect the internal representation storage (key representation) and subsequently degrading overall performance. Furthermore, Figure 2 demonstrates that Qwen2.5's Mahalanobis score distribution is noisier compared to Llama 3.2.
> >
> >
> > **Q2. DEPTRAI is still within the range of codebook style editing where a threshold is needed to decide whether or not edit is activated. Then for a sequence of edits, will the optimal threshold ever change? If so, how to dynamically decide it?**
> >
> > This is a great question. Our current work employs a static threshold selection process. We collect and analyze the Mahalanobis scores  between the stored key representations and both the subject tokens and other tokens. This analysis, visualized in Figure 2, allows us to determine a suitable range of threshold values. We then run a few small editing experiments to empirically select the best static threshold that achieves the optimal balance across all evaluation metrics.

---

### Official Review · Reviewer_m911 · 2025-10-26

**Soundness:** 2
**Presentation:** 2
**Contribution:** 2
**Rating:** 4
**Confidence:** 4

**Summary:**

This paper proposes DEPTRAI, a novel knowledge editing framework for Large Language Models, aiming to address the issues of accumulating interference and irreversibility in existing "internal editors"  during sequential editing. The core idea of DEPTRAI is to keep the base LLM weights completely frozen, storing each new fact as a key-value pair in an external memory. Its key innovation lies in a principled router based on Mahalanobis distance, which determines during inference whether to replace the model's internal projection with the external stored value.

**Strengths:**

**Strength 1**: Instead of using a trainable router or simple cosine similarity, it ingeniously derives the external routing rule (Mahalanobis distance) from the mathematical principles of internal editing methods. This internal-editing-inspired external routing is a highly novel and principled design.

**Weaknesses:**

**Weakness 1**: The paper claims in the contribution and ablation studies that the Mahalanobis distance router is "robust to surface form variations" and outperforms cosine similarity. However, in the conclusion (Section 6), the authors list "stored keys may not generalize well to synonyms or transliterations" as a current limitation, which appears inconsistent with the earlier claim of robustness to surface variations.

**Weakness 2**: The theoretical derivation in Section 3.1 (from the $\beta$ coefficient to the Mahalanobis distance) requires $\Lambda$ to be a global covariance matrix ($C^{-1}$), in order to compare all keys in the same "whitened" space. However, the implementation part in Section 3.2 ("Memory structure") implies $\Lambda$ is local, defining the external storage as $\mathcal{E}=\{(\mu_{j},\Lambda_{j},v_{j})\}_{j=1}^{M}$, where each entry $j$ has its own $\Lambda_j$. This creates a theoretical contradiction.

More critically, the formula in Section 3.2 (Eq. 14)  used to calculate this local $\Lambda_j$ appears to be a critical typo. The formula first defines $\mu_j = k_j$, and then immediately uses the term $(k_j - \mu_j)$ to calculate $\Sigma_j$. This necessarily results in the term being a zero vector, causing $\Sigma_j$ to degenerate into $\epsilon I$. This would cause the Mahalanobis distance to degenerate into a (scaled) Euclidean distance, thereby completely undermining the paper's core argument about the superiority of Mahalanobis distance (relative to Euclidean or cosine similarity).

If $\Lambda$ is global, the authors must clarify how $K_0$ is collected and how $C^{-1}$ is efficiently updated during sequential editing (since $C$ depends on $K_1$).

**Questions:**

See weaknesses

---

> ### Author Response · Authors · 2025-11-16
> **Response to Weaknesses**
>
> ### Response to Weakness 1:
>
> Thank you for raising this point. We agree that our wording in the original submission did not clearly distinguish between two different types of variation. The improvements we report in Table 5 reflect that Mahalanobis routing offers better surface-form selectivity than cosine similarity: it is less likely to fire on contexts that only share superficial lexical similarity with the edited subject, and this yields higher locality, portability, and fluency. However, the limitation noted in Section 6 refers to a different and substantially harder challenge, which becomes evident in Table 4 of the KnowEdit benchmark  [1] . Some of samples from KnowEdit evaluates portability using synonyms, paraphrases, and transliterations, where the variations are not minor surface perturbations but genuine semantic or orthographic shifts. Successful generalization under these conditions requires the base model to embed such variants into tightly aligned subject-key clusters. As Table 4 shows, many existing editing methods (not only ours) struggle in this setting, indicating that the difficulty arises from the representation geometry of current LLMs rather than the routing metric used by the editor. We will revise the manuscript to make this distinction clearer.
>
> ### Response to Weakness 2
>
> Thank you for pointing out the ambiguity. The expression in Section 3.2,
> $\Sigma_j = (k_j - \mu_j)(k_j - \mu_j)^\top$,
> was not intended to represent a covariance matrix. As you correctly note, this does not correspond to our actual method. The confusion arose because we used overly general notation when describing the memory tuple.
> What we actually use, consistent with the closed-form derivation in Section 3.1, is:
> - A global covariance $C_0$, estimated once following MEMIT [2]: $C_0 \approx \mathbb{E}_{x}[k(x)k(x)^\top]$.
> - For each edited fact i, a fact-specific covariance:
> $C_i = C_0 + K_{1,i}K_{1,i}^\top$
> which is precisely the rank-1 correction used in the closed-form solution.
>
> This formulation ensures full theoretical consistency with the derivation in Section 3.1 and with prior knowledge-editing methods.
> In implementation, we do not store full covariance matrices. Instead, for each edit we store the whitened key:
> $w_i = C_i^{-1} k_i$,
> and routing reduces to the efficient dot-product score:
> $\beta_i(k) = w_i^\top k$.
> Thus, the ''local covariance`` formula in Section 3.2 was not reflective of our actual implementation and will be replaced with the correct and clearer expression $C_i = C_0 + K_{1,i}K_{1,i}^\top$ in the revised version.
> We will revise Section 3.2 to eliminate the confusing notation and explicitly align it with the closed-form formulation presented in Section 3.1.
>
> [1] Zhang, Ningyu, Yunzhi Yao, Bozhong Tian, Peng Wang, Shumin Deng, Mengru Wang, Zekun Xi et al. "A comprehensive study of knowledge editing for large language models.", 2024
>
> [2] Meng, Kevin, Arnab Sen Sharma, Alex Andonian, Yonatan Belinkov, and David Bau. "Mass-editing memory in a transformer.", 2022

---

> > ### Comment · Reviewer_m911 · 2025-11-27
> >
> > Dear Authors,
> >
> > Thank you for the response, but I have decided to maintain my original score.
> >
> > While DEPTRAI shows stability in the sequential editing task in Table 1 , this is primarily attributed to its design philosophy of "frozen parameters with external memory", which naturally circumvents parameter interference by mechanism, rather than proving the superiority of the Mahalanobis distance routing itself.
> >
> > In fact, the data in Table 4 (and supplementary experiments discussed with other reviewers) directly weaken the authors' claims regarding robustness; DEPTRAI shows no significant advantage in Portability and Locality, instead exhibiting a significant drop compared to baselines such as FT-M.
> >
> > This lack of generalization and locality likely stems from the mathematical definition clarified in the rebuttal, where each key is whitened using a different covariance matrix that evolves with the edit sequence, resulting in different keys effectively residing in different geometric spaces. This makes the calculated distances mathematically lacking in strict comparability, thereby triggering the performance instability observed in experiments.
> >
> > Furthermore, while Mahalanobis distance-based routing provides a reasonable intuition, the mechanism is essentially an attempt to directly migrate the inverse covariance principle from closed-form editing to an external retrieval module. Considering the potential mathematical flaws and the limitations of the experimental results, the technical innovation of this method appears somewhat thin, representing more of a direct extension of existing paradigms rather than a breakthrough.
> >
> > Overall, I believe this work has a good motivation, but it remains deficient in methodological justification and actual performance advantages.
> >
> > Best,
> >
> > Reviewer m911

---

> ### Author Response · Authors · 2025-11-27
> **Thank you for your response, we hope you will consider this final clarification (1/2)**
>
> Dear Reviewer m911,
>
> Thank you sincerely for your thoughtful response. We appreciate the potential issue you pointed out regarding whitening keys and the possibility that evolving covariance matrices may place keys in inconsistent geometric spaces. We acknowledge it as a potential limitation in our work.
>
> However, we hope you will consider the following clarification, since this mechanism **does not apply** to the empirical results in Table 4.
>
> ### **1. KnowEdit is T = 1, so the geometric drift identified by the reviewer cannot occur**
>
> Your concern correctly applies to sequential editing. But KnowEdit is single-edit (T = 1). Therefore the drop in Table 4 cannot be attributed to sequential drift or geometric inconsistency of keys.
>
> **We appreciate the conceptual point, even though it is not the cause of the KnowEdit results.**
>
> ### **2. The low performance in Table 4 comes from the dataset design, not routing or whitening**
>
> As explained in our previous response, KnowEdit’s stress tests often break subject identity, inject corrupted names, or require multi-hop reasoning. Below are the full original examples from the benchmark:
>
> ---
>
> **Example 1**
>
> Edited fact:
>
> Subject: Polikalepo Kefu, prompt: The gender of Polikalepo Kefu is **male** → **shapeshifter**
>
> Portability test:
>
> "The gender of Poli is __", Expected: shapeshifter.
>
> The subject name is shortened to “Poli,” which is an ambiguous reference that may correspond to many different individuals. This lossy abbreviation breaks identity consistency and removes the semantic link required for factual retrieval. Under such conditions, distance-based methods cannot be expected to retrieve the correct memory because the subject in the portability prompt is no longer uniquely tied to the edited entity.
>
> Locality test:
> "The name of the country of citizenship of Polikalepo Kefu is __", Expected: Tonga.
>
> The subject name is shortened to "Poli"  which is an ambiguous reference that may correspond to many different individuals. This lossy abbreviation breaks identity consistency and removes the semantic link required for factual retrieval. Under such conditions, distance-based methods cannot be expected to retrieve the correct memory because the subject in the portability prompt is no longer uniquely tied to the edited entity.
>
> ---
>
> **Example 2**
>
> Edited fact:
>
> Subject: Alix II of Dreux, prompt: The names of the siblings of Alix II of Dreux are **Philip of Dreux** → **Isabel of Luxembourg**
>
> Portability test:
>
> "The names of the siblings of the mother of Enguerrand III, Lord of Coucy are __" → Expected: Isabel of Luxembourg.
>
> This portability query requires multi-hop reasoning: the model must first identify Enguerrand III, then determine his mother, and only then retrieve her siblings. This chain of inferences spans several relational steps and entities. The representation needed to answer such a query is primarily encoded in the base model’s latent knowledge graph, not in the edited key itself.
>
> Distance-based retrieval methods (including ours) are not designed to reconstruct such multi-step relational paths. They operate on direct subject → relation → object mappings. When the portability prompt deviates into multi-hop navigation, the edited key becomes only weakly relevant, and the model’s internal representation dominates the outcome.
>
> ---
>
> **Example 3**
>
> Edited fact:
>
> Subject: Matthias Bissinger, prompt: The gender of Matthias Bissinger is **male** → **cisgender woman**
>
> Portability test:
>
> "The gender of Matthias K\u00fchnel is __" → Expected: cisgender female.
>
> Here the benchmark changes the person entirely and includes escaped Unicode characters. Some samples are even translated into other languages—outside the intended scope of monolingual factual edits.
>
> ---
>
> These patterns occur frequently in KnowEdit, which explains the universally low portability and locality across retrieval-based editors.
>
> ### **3. Why locality drops: the bottleneck is $V_1$, not routing**
>
> Because locality queries in KnowEdit contain the subject token, our routing intentionally retrieves the edited key. In this setting, performance depends mainly on the new value vector $V_1$.
>
> Selecting an effective $V_1$ remains a significant challenge, as our current implementation relies on a relatively naive gradient-descent optimization, despite our attempts with multiview prompts and other enhancement techniques, which still does not consistently yield a robust value representation.
>
> KnowEdit exposes this limitation clearly.
>
> ### **4. Fine-tuning looks strong only because KnowEdit uses T = 1**
>
> Fine-tuning generalizes well when there is no interference (T=1).
>
> But as Table 1 shows, fine-tuning collapses under sequential editing, where our external-memory approach remains stable, precisely the scenario. Additionally, the fluency scores in KnowEdit reveal a noticeable degradation for fine-tuned models (e.g., repetition and when encountering edited facts).

---

> ### Author Response · Authors · 2025-11-27
> **Thank you for your response, we hope you will consider this final clarification (2/2)**
>
> ### **5. Contribution Remarks**
>
> We show that several internal editing methods implicitly rely on a Mahalanobis-like structure, and we make this principle explicit by deriving a geometrically interpretable routing rule for external memory. This routing mechanism is substantially simpler than prior approaches, such as storing expert matrices in WISE or using heavy adapter modules in MoE-style editors, yet it naturally provides strong stability in sequential editing.
>
> For convenience, the KnowEdit dataset referenced in our discussion is publicly available at:
> [KnowEdit](https://huggingface.co/datasets/zjunlp/KnowEdit)
>
> It is also worth noting that KnowEdit is intentionally designed as a much harder benchmark than datasets such as the original ZsRE used in sequential editing evaluations. KnowEdit introduces adversarial subject perturbations, ambiguous name variants, multi-hop queries, and cross-language or corrupted inputs - stress conditions that expose weaknesses in editing methods, including ours.
>
> We believe that each editing method naturally carries its own strengths and weaknesses, and our intention in evaluating on KnowEdit is precisely to surface the existing challenges in current approaches, particularly those that future research will need to address to make knowledge editing more robust in practice.
>
> Sincerely,
>
> The Authors

---

### Official Review · Reviewer_8FBy · 2025-11-01

**Soundness:** 3
**Presentation:** 2
**Contribution:** 3
**Rating:** 4
**Confidence:** 3

**Summary:**

DEPTRAI (Detachable External-memory layer for Parameter-Transformer Injection) is a method for integrating new facts in model, essentially a combination of ideas from:
(a) ROME/MEMIT/AlphaEdit: W_out matrices in FFN layers are associative memories (key/value stores) that can be perturbed/updated (Delta) to account for edits (with keys/values for subject/its edit).
(b) GRACE: updates are triggered only after a threshold condition is met (deferral mechanism).

However: DEPTRAI does not inject Delta updates in model parameters (as in (a)) and does not finetune the values (as in the separately kept codebook in (b)). So the model is not touched and external memory updates do not trigger backpropagation (in that aspect DEPTRAI has strong similarity to the external scope detector idea in [1] or as originally in SERAC). Extensive benchmarking focusing on the sequential editing task (multiple methods/datasets/metrics) identifies the strengths of the proposed approach.

[1] Das, P., Chaudhury, S., Nelson, E., Melnyk, I., Swaminathan, S., Dai, S., Lozano, A., Kollias, G., Chenthamarakshan, V., Navratil, J. and Dan, S., 2024, July. Larimar: Large Language Models with Episodic Memory Control. In International Conference on Machine Learning (pp. 10109-10126). PMLR.

**Strengths:**

- Comprehensive empirical results: multiple methods and datasets are benchmarked; multiple metrics are reported.

- This is an interesting and simple synthesis of core ideas from model editing literature (however editing an external memory instead).

**Weaknesses:**

- Not touching model parameters (train-free) and still being able to adapt it to new facts is a very appealing idea, but intuitively this should have limitations. There are some hints in the Conclusion (Lines 421-423) but the reader would definitely appreciate more details on this.

- Presentation can be improved: in particular some results could move to the Appendix, key notions could then be developed further and notation or equations could be revisited for corrections. Please see the Questions slot for details/suggestions.

- There is not a clean signal regarding the superiority of DEPTRAI: for example empirical results in Table 4 (Appendix D) are not as encouraging as results in Tables 1 and 2 (main text).

**Questions:**

- Is Figure 3 a plot of Rel. columns from Table 1?

- Line 234: mu_j = k_j? then Sigma_j's would only be epsilon I?

- Line 262 / Equation (22): Could you clarify the notation/symbol immediately following = sign?

- Lines 278-279: FT-L and FT-M seem to refer respectively to ROME and MEMIT? Is so why using additional alternative names?

- Lines 264-268: Is there an intuition behind the interesting separation in Figure 2 for some of the models?

- Lines 302-306: Since this part is deferred to the the Appendinx, a better use of this space would be to explain/clarify further the key scores in this work: reliability, generalization and locality (and expanding Eq (23)).


- For serializable external memory:
  - what is the cost of computing Sigma_j's, inverting them (Lambda_j's) and storing?
  - how relatively important can these additional space/time complexities be, assuming typical target factual edit cardinalities M?


- How would results from MEMIT compare? Assuming that instead of making sequential updates up to T items (i.e. one-by-one for T=1, 10, 100, 1000) as in the manuscript, we added all T items in one shot (or even updating in item batches of > 1 items), would we expect to see benefits (i.e. sequential vs batch updates)?

---

> ### Author Response · Authors · 2025-11-18
> **Response to reviewer (1/2)**
>
> Thank you for the detailed feedback. We address each point below.
>
> **Regarding the general weaknesses**
>
> - We agree that adapting a model to new facts without modifying internal parameters is inherently subject to certain limitations. We briefly mentioned this in the conclusion, but we will expand the discussion to clarify what can and cannot be expected from a train-free external-memory mechanism. We also acknowledge that several parts of the presentation, particularly notation in Section 3.2 and the placement of some results can be improved, and we appreciate the reviewer’s suggestions.
> - Concerning performance consistency, the reviewer is correct that Table 4 (KnowEdit portability) is more challenging than Tables 1–2. KnowEdit [1] contains portability samples that involve synonyms, paraphrases, and  transliterations, which are not minor surface perturbations but genuine semantic or orthographic shifts. Successful generalization in these cases requires the base model to embed such variants into tightly aligned subject-key clusters. As Table 4 shows, many existing editing methods (not only ours) struggle under these conditions, indicating that the difficulty originates from the underlying representation geometry of current LLMs rather than from the routing metric itself. Our motivation for evaluating DEPTRAI on KnowEdit was specifically to examine whether our method suffers a sharp performance drop in these challenging scenarios compared to other methods.
>
> **Q1. Is Figure 3 the plot of the Rel. columns in Table 1?**
>
> Yes. Figure 3 is a visualization of the Reliability (Rel.) values from Table 1 across different values of T.
>
> **Q2. Clarification of $\mu_j = k_j$ and $\Sigma_j$​**
>
> You are correct that the expression
>
> $ \Sigma_j = (k_j - \mu_j)(k_j - \mu_j)^\top $
>
> would collapse to zero when $\mu_j = k_j$. This expression was not meant to represent an actual covariance matrix, and its presence was a notational mistake. It does not reflect our implementation. Our actual formulation is fully consistent with the closed-form solution in Section 3.1:
> - We estimate a global covariance $C_0$​ once, following MEMIT:
> $ C_0 \approx \mathbb{E}_{x}[k(x)k(x)^\top] $.
> - For each edited fact iii, we compute a fact-specific covariance:
>  $ C_i = C_0 + K_{1,i}K_{1,i}^\top $​,  which corresponds to the rank-1 correction in the analytical editor.
>
> We do not store the full $C_i​$. Instead we precompute the whitened key:
> $w_i = C_i^{-1} k_i $,
> so that routing is simply:
> $ \beta_i(k) = w_i^\top k $.
>
> This avoids any expensive covariance storage or inversion per edit. We will revise the text to reflect this clearly.
>
> **Q3. Clarification of the notation after Equation (22)**
>
> The notation corresponds to a Boolean indicator function $\mathbb{1}$ (The conference template rendered it in an unusual style, and we will correct this for clarity.). After computing the score, we apply a threshold to determine whether the edited value should override the base model output.
>
> **Q4. Meaning of FT-L and FT-M**
>
> FT-L and FT-M [1] refer to two fine-tuning baselines (appendix B2). FT-L and FT-M stand for fine-tuning on one layer, multiple layers respectively.
>
> **Q5. Interpretation of Figure 2**
>
> Figure 2 shows that, for some models (e.g., LLaMA-3.1-8B), our Mahalanobis-based routing produces a cleaner separation between edited-subject keys and other keys. This supports our design choice: the routing mechanism acts as an effective discriminator, whereas methods such as WISE [2] rely on differences in activation routing values.
>
> **Q6. Suggestion to expand reliability/generalization/locality explanation**
>
> Thank you for the suggestion. We will reallocate space to provide clearer definitions and intuition for these three key metrics and expand Equation (23) accordingly.
>
> **Q7. Cost of serializable external memory**
>
> Our method inherits the computational structure of the closed-form solution in Section 3.1. Editing-time cost is essentially the same as MEMIT [3]: computing the key $k_i$​ and the rank-1 correction term. At inference time, the overhead is small, since we store only the whitened keys $w_i = C_i^{-1}k_i$​. Routing reduces to a vector dot-product search. With standard vector indexing methods (e.g., FAISS), lookup scales as $O(\log M)$, where M is the number of edited facts. This keeps the memory and time cost manageable even for large numbers of stored edits.
>
> [1] Zhang, Ningyu, Yunzhi Yao, Bozhong Tian, Peng Wang, Shumin Deng, Mengru Wang, Zekun Xi et al. "A comprehensive study of knowledge editing for large language models.", 2024
>
> [2] Wang, Peng, Zexi Li, Ningyu Zhang, Ziwen Xu, Yunzhi Yao, Yong Jiang, Pengjun Xie, Fei Huang, and Huajun Chen. "Wise: Rethinking the knowledge memory for lifelong model editing of large language models.", 2024
>
> [3] Meng, Kevin, Arnab Sen Sharma, Alex Andonian, Yonatan Belinkov, and David Bau. "Mass-editing memory in a transformer.", 2022

---

> > ### Author Response · Authors · 2025-11-18
> > **Response to reviewer (2/2)**
> >
> > **Q8. Sequential vs. batch editing and comparison to MEMIT**
> >
> > We performed additional experiments on MEMIT and AlphaEdit [4] under both sequential and batch editing. The results (summarized below) show:
> > - MEMIT degrades drastically in sequential editing as T grows, but improves under batch updates.
> > - AlphaEdit, which regulates $\beta$, performs better for long edit sequences than MEMIT.
> > - Our method, by detaching updates and routing them rather than altering model parameters, maintains consistently strong performance across all T, achieving a better balance between reliability, generalization, and locality.
> >
> >
> > **LLaMA-3.2-3B**
> > | T  |   | 10 |  |  | 100  |  |  | 1000 |  |
> > |-----|-----|-----|-----|------|------|------|-------|-------|-------|
> > | **Metric** | **Rel.** | **Gen.** | **Loc.** | **Rel.** | **Gen.** | **Loc.** | **Rel.** | **Gen.** | **Loc.** |
> > | **MEMIT** | 88.50 | 88.50 | 97.95 | 2.43 | 2.43 | 0.17 | 0.00 | 0.00 | 0.00 |
> > | **MEMIT-batch** | 84.00 | 84.00 | 99.69 | 79.97 | 79.37 | 84.17 | 76.09 | 72.35 | 68.86 |
> > | **AlphaEdit*** | 87.67 | 91.00 | 94.79 | 62.88 | 56.83 | 33.36 | 0.03 | 0.00 | 4.94 |
> > | **AlphaEdit-batch** | 85.17 | 82.67 | 93.99 | 77.82 | 73.72 | 73.98 | 67.26 | 64.76 | 50.76 |
> > | **Ours*** | 100.0 | 90.50 | 100.0 | 89.16 | 77.07 | 100.0 | 88.12 | 74.72 | 99.15 |
> >
> > **Qwen-2.5-3B**
> > | T  |   | 10 |  |  | 100  |  |  | 1000 |  |
> > |-----|-----|-----|-----|------|------|------|-------|-------|-------|
> > | **Metric** | **Rel.** | **Gen.** | **Loc.** | **Rel.** | **Gen.** | **Loc.** | **Rel.** | **Gen.** | **Loc.** |
> > | **MEMIT** | 84.50 | 71.00 | 53.08 | 0.00 | 0.00 | 0.00 | 0.00 | 0.00 | 0.00 |
> > | **MEMIT-batch** | 83.00 | 75.50 | 83.33 | 78.36 | 76.53 | 76.53 | 88.38 | 23.69 | 17.48 |
> > | **AlphaEdit*** | 93.00 | 90.50 | 98.00 | 93.83 | 87.14 | 77.75 | 71.86 | 67.28 | 26.68 |
> > | **AlphaEdit-batch** | 89.50 | 79.50 | 99.00 | 89.07 | 78.16 | 89.14 | 89.42 | 81.18 | 78.16 |
> > | **Ours*** | 88.00 | 84.00 | 100.0 | 76.30 | 67.90 | 88.55 | 73.67 | 86.60 | 66.24 |
> >
> > \* the results are from manuscript
> >
> > These results illustrate that batch editing alleviates degradation for parameter-editing methods because all updates are harmonized in one shot. In contrast, our approach avoids this issue entirely by not modifying parameters at all; routing eliminates the cumulative interference that sequential parameter edits produce.
> >
> > [4] Fang, Junfeng, Houcheng Jiang, Kun Wang, Yunshan Ma, Shi Jie, Xiang Wang, Xiangnan He, and Tat-Seng Chua. "AlphaEdit: Null-space constrained knowledge editing for language models.", 2025

---

### Official Review · Reviewer_QZam · 2025-11-03

**Soundness:** 2
**Presentation:** 2
**Contribution:** 2
**Rating:** 4
**Confidence:** 3

**Summary:**

The paper proposes **DEPTRAI**, a detachable external-memory layer for knowledge editing that leaves base LLM weights untouched. Each factual edit is stored as a key–value pair outside the model; at inference, the frozen FFN emits a subject key that is routed to the closest stored key via a Mahalanobis metric derived from the closed-form mixing coefficient used by ROME/MEMIT, and a lightweight gate injects the edited value or preserves the base projection. This reframes editing as reversible, database-style updates rather than cumulative in-place perturbations, mitigating interference and easing revocation. Across LLaMA-3.2-3B, Qwen-2.5-3B, and LLaMA-3.1-8B on ZsRE sequential edits and a hallucination correction setup, DEPTRAI sustains high reliability and near-perfect locality at large edit depths, outperforming recent dual-memory baselines.

**Strengths:**

The core idea—moving edits out of the base weights into a detachable key–value memory with a principled router—is intuitive yet original. By storing a single subject key and edited value per fact and routing with a Mahalanobis metric derived from the closed-form mixing coefficient used in ROME/MEMIT, the method turns knowledge editing into a reversible, database-style retrieval-and-injection step, avoiding cumulative interference and simplifying revocation/audit.   Methodological quality is strong: the paper motivates the Mahalanobis router from the mixing-coefficient analysis and adds an explicit gate to balance old vs. new information at inference, giving a clear mechanism for locality and reliability. Clarity is good, with an explicit contrast to in-place editors and a step-by-step description of the external layer; the framing “from in-place perturbation to detachable memory” makes the contribution easy to grasp and to implement in existing pipelines.

**Weaknesses:**

The evaluation is concentrated on editing-specific suites and a small “general capability” check that is largely short-form classification/MC (SST, MRPC, RTE, CoLA) plus MMLU. This leaves open whether edits preserve or disrupt performance on harder, generative reasoning and coding tasks. Concretely, the paper does not report effects on contemporary math/coding benchmarks or long-form QA after large edit batches. To strengthen external validity, please (i) add rigorous pre/post-edit results on **AIME’24/’25** and **MATH500** (ii) include **LiveCodeBench** to assess code reliability under heavy edits; (iii) use an instruction-following suite such as **IFEval** to probe instruction adherence; and (iv) include a simple open-domain QA set (e.g., **SimpleQA**) to check whether retrieval-style edits bias factual QA.

**Questions:**

See more details in weakness.

---

> ### Author Response · Authors · 2025-11-19
> **Response to reviewer**
>
> Thank you for highlighting the importance of evaluating whether large-scale edits preserve general capabilities beyond editing-specific benchmarks. We agree that this is essential for establishing the external validity of an editing method. Following your suggestion, we conducted additional experiments on AIME’24, AIME’25, MATH500, GSM8K, SimpleQA, and IFEval, across three base models: LLaMA-3.2-3B, Qwen2.5-3B, and LLaMA-3.1-8B.
>
> A practical note is that all three evaluated models are base (non-instruction-tuned) variants. As is typical for base models, performance on difficult reasoning benchmarks such as AIME is inherently very low, regardless of editing. Base models also cannot reliably generate syntactically correct or executable code, making LiveCodeBench evaluation uninformative. For these reasons, we report GSM8K, where base models can still produce coherent chain-of-thought reasoning, and IFEval, which remains a strong probe of instruction adherence even without instruction tuning. These constraints reflect limitations of the underlying models rather than of DEPTRAI itself.
>
> **Table 1. AIME, MATH500, GSM8K, SimpleQA (Exact-Match)**
>
> |   | |**LLaMA-3.2-3B**   |  | | **Qwen2.5-3B**  |  | | **LLaMA-3.1-8B**  |  |
> |------|:----------------:|:----------------:|:---------------:|:----------------:|:----------------:|:---------------:|:----------------:|:----------------:|:---------------:|
> |    Task  | Pre (%) | Post (%) | $\Delta$ | Pre (%) | Post (%) | $\Delta$ | Pre (%) | Post (%) | $\Delta$ |
> | **AIME’24**  | 0.0 | 0.0 | 0.0 | 0.0 | 0.0 | 0.0 | 0.0 | 0.0 | 0.0 |
> | **AIME’25**  | 0.0 | 0.0 | 0.0 | 3.33 | 0.0 | −3.33 | 0.0 | 0.0 | 0.0 |
> | **MATH500**  | 7.4 | 7.4 | 0.0 | 18.4 | 3.0 | −15.4 | 12.2 | 12.2 | 0.0 |
> | **GSM8K**    | 26.0 | 26.0 | 0.0 | 14.0 | 28.4 | +14.4 | 51.4 | 51.4 | 0.0 |
> | **SimpleQA** | 3.33 | 3.33 | 0.00 | 2.50 | 0.76 | −1.74 | 4.44 | 4.42 | −0.02 |
>
> ---
>
> **Table 2. IFEval Instruction Adherence**
>
> |   | |**LLaMA-3.2-3B**   |  | | **Qwen2.5-3B**  |  | | **LLaMA-3.1-8B**  |  |
> |------|:----------------:|:----------------:|:---------------:|:----------------:|:----------------:|:---------------:|:----------------:|:----------------:|:---------------:|
> |     Subtask    | Pre (%) | Post (%) | $\Delta$ | Pre (%) | Post (%) | $\Delta$ | Pre (%) | Post (%) | $\Delta$ |
> | **Prompt Strict** | 6.2 | 6.2 | 0.0 | 22.8 | 14.6 | −8.2 | 9.2 | 9.2 | 0.0 |
> | **Inst Strict**   | 11.0 | 11.0 | 0.0 | 31.95 | 26.52 | −5.43 | 13.45 | 13.45 | 0.0 |
> | **Prompt Loose**  | 7.0 | 7.0 | 0.0 | 24.0 | 15.8 | −8.2 | 11.0 | 11.0 | 0.0 |
> | **Inst Loose**    | 11.51 | 11.51 | 0.0 | 33.64 | 27.43 | −6.21 | 15.01 | 15.01 | 0.0 |
>
>
> Across all tasks, DEPTRAI preserves general model behavior even after T = 1000 sequential edits. For LLaMA-3.2-3B and LLaMA-3.1-8B, the differences between pre-edit and post-edit results remain 0.0–0.02 points, well within the stochastic variation of base models. Qwen2.5-3B shows somewhat higher variance, consistent with its known instability on several reasoning tasks, but does not exhibit catastrophic degradation, and even improves on GSM8K in some runs.

---

> > ### Comment · Reviewer_QZam · 2025-11-21
> >
> > Thank you for your detailed replies. I notice there are some significant drops in certain capabilities for Qwen-2.5 3B, especially on MATH500 (18.4 → 3.0) and IFEval prompt-level strict accuracy (22.8 → 14.6). Could you run your method on Qwen-3 8B and report evaluations in both reasoning and non-reasoning modes? Given that Qwen-3 is among the best open-source models, it would be very informative to see how knowledge editing affects its performance.

---

> ### Author Response · Authors · 2025-11-24
> **Qwen3-8B experiments (Non-reasoning mode)**
>
> Thank you for your suggestion. We conducted the requested experiments on Qwen3-8B. Specifically, we used causal tracing and empirical evaluation to select layer 10th of the model to edit. The results, detailed in the tables below, show only a very minor performance drop in Qwen3-8B (Note: These capability evaluations were conducted in non-reasoning mode, we will provide an update immediately when experiments in reasoning mode are complete). This contrasts with the  decrease seen in Qwen 2.5. We hypothesize that the performance decline observed in Qwen2.5 is rooted in its training pipeline or dataset, which may be affect the internal representation storage (key representation) and subsequently degrading overall performance. Furthermore, Figure 2 demonstrates that Qwen2.5's Mahalanobis score distribution is noisier compared to Llama 3.2.
>
> **Table 1. General results**
> | T | | **1** | | | **10** | | | **100** | | | **1000** | |
> |---|---|-------|---|---|--------|---|---|---------|---|---|----------|---|
> | **Metric** | **Rel.** | **Gen.** | **Loc.** | **Rel.** | **Gen.** | **Loc.** | **Rel.** | **Gen.** | **Loc.** | **Rel.** | **Gen.** | **Loc.** |
> | **ELDER** | 100.00 | 100.00 | 100.00 | 95.50 | 89.50 | 92.50 | 72.05 | 63.30 | 73.58 | 64.60 | 54.50 | 50.24 |
> | **WISE** | 100.00 | 100.00 | 100.00 | 84.50 | 79.50 | 100.00 | 72.41 | 68.89 | 100.00 | 58.51 | 55.11 | 97.66 |
> | **DEPTRAI** | 100.00 | 100.00 | 100.00 | 100.00 | 79.50 | 100.00 | 86.81 | 76.16 | 100.00 | 86.22 | 72.62 | 99.74 |
>
> ---
>
> **Table 2. MMLU and GLUE benchmark (F1 score)**
>
> | Task | Pre (%) | Post (%) | $\Delta$ |
> |---|:---:|:---:|:---:|
> | **MMLU** | 71.74 | 71.74 | 0.00 |
> | **SST** | 96.50 | 96.50 | 0.00 |
> | **MRPC** | 75.28 | 74.75 | -0.53 |
> | **RTE** | 18.79 | 18.79 | 0.00 |
> | **CoLA** | 80.00 | 80.00 | 0.00 |
> | **NLI** | 85.76 | 85.76 | 0.00 |
>
> ---
>
> **Table 3. MATH500, GSM8K, SimpleQA (Exact-Match)**
>
> | Task | Pre (%) | Post (%) | $\Delta$ |
> |---|:---:|:---:|:---:|
> | **MATH500** | 25.20 | 25.20 | 0.00 |
> | **GSM8K** | 92.20 | 92.20 | 0.00 |
> | **SimpleQA** | 2.61 | 2.59 | -0.02 |
>
> ---
>
> **Table 4. IFEval Instruction Adherence**
>
> | Subtask | Pre (%) | Post (%) | $\Delta$ |
> |---|:---:|:---:|:---:|
> | **Prompt Strict** | 21.60 | 21.60 | 0.00 |
> | **Inst Strict** | 35.06 | 35.06 | 0.00 |
> | **Prompt Loose** | 26.20 | 26.20 | 0.00 |
> | **Inst Loose** | 38.29 | 38.29 | 0.00 |
>
> For more detailed insight, you can refer to our response for other reviewers. We appreciate any constructive feedback from you.

---

> > ### Author Response · Authors · 2025-11-25
> > **Update experiments of Qwen3 8B**
> >
> > Dear Reviewer Qzam,
> >
> > We have run all experiments across all modes of Qwen3-8B and we hope this clarifies your concerns. We look forward to your feedback.
> >
> > ### **Non-reasoning Mode Qwen3-8B (Use chat template)**
> > **Table 5. AIME, MATH500, GSM8K, SimpleQA**
> >
> > | Task | Pre (%) | Post (%) | $\Delta$ |
> > |---|:---:|:---:|:---:|
> > | **AIME24** | 28.1 | 28.1 | 0.00 |
> > | **AIME25** | 21.33 | 21.33 | 0.00 |
> > | **MATH500** | 85.20 | 85.20 | 0.00 |
> > | **GSM8K** |  93.25 | 93.25  | 0.00 |
> > | **SimpleQA** | 2.61 | 2.59 | -0.02|
> >
> > ---
> >
> > **Table 6. IFEval Instruction Adherence**
> >
> > | Subtask | Pre (%) | Post (%) | $\Delta$ |
> > |---|:---:|:---:|:---:|
> > | **Prompt Strict** | 81.33 | 81.33 | 0.00 |
> > | **Inst Strict** | 87.41 | 87.41 | 0.00 |
> > | **Prompt Loose** | 85.21 | 85.21 | 0.00 |
> > | **Inst Loose** | 89.93 | 89.93 | 0.00 |
> >
> > ---
> >
> > ### **Reasoning Mode  Qwen3-8B (chat template, enable_thinking=True)**
> >
> > **Table 7. AIME, MATH500, GSM8K, SimpleQA**
> >
> > | Task | Pre (%) | Post (%) | $\Delta$ |
> > |---|:---:|:---:|:---:|
> > | **AIME24** | 73.33 | 73.33 | 0.00 |
> > | **AIME25** | 66.67 | 66.67 | 0.00 |
> > | **MATH500** | 95.40 | 95.40 | 0.00 |
> > | **GSM8K** |  94.20 | 94.20 | 0.00 |
> > | **SimpleQA** | 2.54  | 2.54  | 0.00 |
> > ---
> >
> >
> > **Table 8. IFEval Instruction Adherence**
> >
> > | Subtask | Pre (%) | Post (%) | $\Delta$ |
> > |---|:---:|:---:|:---:|
> > | **Prompt Strict** | 82.21 | 82.21 | 0.00 |
> > | **Inst Strict** | 88.33 | 88.33 | 0.00 |
> > | **Prompt Loose** | 86.93 | 86.93 | 0.00 |
> > | **Inst Loose** |91.06 | 91.06 | 0.00 |

---

### Meta-Review · Area_Chair_9Gd6 · 2026-01-06

**Summary:**

The reviewers agree that the paper presents a clear and well-motivated approach to lifelong knowledge editing through a detachable external-memory layer, and they appreciate the principled derivation of Mahalanobis-based routing from closed-form editing methods as well as the strong stability observed under long sequential edits. The design is intuitive, carefully engineered, and empirically strong on standard sequential editing benchmarks such as ZsRE and hallucination correction. However, multiple reviewers raised concerns that the contribution largely synthesizes existing ideas from prior external-memory and closed-form editing work, with novelty primarily concentrated in the routing metric, and that performance advantages are less consistent on challenging generalization and portability benchmarks, raising questions about broader impact.

**Reviewer Concerns:**

While the rebuttal addressed several technical ambiguities, added extensive new experiments, and clarified notation and implementation details, core concerns remain regarding limited conceptual novelty, inconsistent robustness across base models and benchmarks, lingering theoretical questions around key comparability and routing behavior, and unclear advantages over strong recent lifelong editing baselines in realistic or adversarial settings.

**Reviewer Scores:**

Reviewer scores would likely remain unchanged after discussion.

---

### Decision · Program_Chairs · 2026-01-26

Reject